# FAT10 inhibits TRIM21 to down-regulate antiviral type-I interferon secretion

Kritika Saxena[1], Katharina Inholz[1], Michael Basler[1,2] , Annette Aichem[1,2]

The ubiquitin-like modifier FAT10 is upregulated under pro-inflammatory conditions, targets its substrates for proteasomal degradation and functions as a negative regulator of the type-I IFN response. Influenza A virus infection upregulates the production of type-I IFN and the expression of the E3 ligase TRIM21, which regulates type-I IFN production in a positive feedback manner. In this study, we show that FAT10 becomes covalently conjugated to TRIM21 and that this targets TRIM21 for proteasomal degradation. We further show that the coiled-coil and PRYSPRY domains of TRIM21 and the C-terminal diglycine motif of FAT10 are important for the TRIM21-FAT10 interaction. Moreover, upon influenza A virus infection and in the presence of FAT10 the total ubiquitination of TRIM21 is reduced and our data reveal that the FAT10-mediated degradation of TRIM21 diminishes IFNβ production. Overall, this study provides strong evidence that FAT10 down-regulates the antiviral type-I IFN production by modulating additional molecules of the RIG-I signaling pathway besides the already published OTUB1. In addition, we elucidate a novel mechanism of FAT10-mediated proteasomal degradation of TRIM21 that regulates its stability.

## Introduction

Pattern recognition receptors (PRRs) are the molecular sensors of invading pathogens and are important for the generation of co-ordinated innate immune responses (Takeuchi & Akira, 2010). They recognize pathogen-associated molecular patterns of bacteria, fungi, viruses, and other pathogens as well as self-components generated as a result of acute injury due to infection, so-called damage-associated molecular patterns (Takeuchi & Akira, 2010). Retinoic acid inducible gene-I (RIG-I)-like receptors (RLRs) are cytosolic PRRs involved in sensing double-stranded RNA (dsRNA) viruses and induce the production of type-I IFNs (Kato et al, 2005). For example, RIG-I binds to short dsRNA having 5' triphosphate, whereas melanoma differentiation factor 5 (MDA5) recognizes long dsRNA. Influenza A virus (IAV) belongs to the Orthomyxoviridae family and contains a negative-sense single-stranded RNA genome (Malik & Zhou, 2020). RIG-I is the cytoplasmic sensor of IAV RNA which binds to its 5'triphosphate end (Yoneyama et al, 2004; Malik & Zhou, 2020). The binding of RNA to RIG-I activates a RIG-I-mediated signaling cascade resulting in the upregulation of type-I IFN production (Akira et al, 2006).

Tripartite motif (TRIM) proteins belong to the largest subfamily of E3 ligases (Li et al, 2008) and are expressed in all multicellular organisms. More than 80 TRIMs have been reported in humans (Di Rienzo et al, 2020; Wang & Ning, 2021). The majority of the TRIM proteins contain three relatively conserved N-terminal domains (RBCC): a really interesting new gene (RING) domain that has E3 ligase activity, a B-box domain (B-box), which regulates the olig-omerization of TRIM proteins, and a coiled-coil (CC) domain which is important for dimerization of TRIM proteins (Shen et al, 2021). The C-terminus of the TRIM proteins contains distinct domains that facilitate substrate recognition, and protein-protein interaction, a few of them exhibit enzymatic activity, or bind nucleic acids (Ozato et al, 2008; Sardiello et al, 2008; Kawai & Akira, 2011; van Gent et al, 2018; Shen et al, 2021). Over the past decades, TRIM proteins have been reported to modulate the PRR-mediated antiviral innate immune response (McNab et al, 2011; Shen et al, 2021). Many of the TRIM proteins are expressed in response to IFNs and are involved in regulating type-I IFN production (Ozato et al, 2008; Hatakeyama, 2017; van Gent et al, 2018).

TRIM21 (also known as Ro52) has been shown to play an important role in the innate immune response, particularly during viral infections (Oke & Wahren-Herlenius, 2012), and its expression is induced by type-I IFNs (Bottermann & James, 2018). However, the exact role of TRIM21 in the antiviral immune response is not yet completely understood because there are contradictory studies showing positive as well as negative effects of TRIM21. For example, in the absence of TRIM21, the innate immune response towards an RNA virus is severely disabled (Foss et al, 2019) and therefore TRIM21 has been classified as a positive regulator of type-I IFN secretion, stabilizing or activating several molecules of the type-I IFN cascade (Li et al, 2023). On the other hand, TRIM21 has been described to be a negative regulator of the innate immune

[1]Department of Biology, Division of Immunology, University of Konstanz, Konstanz, Germany [2]Biotechnology Institute Thurgau at the University of Konstanz, Kreuzlingen, Switzerland

Correspondence: Annette.Aichem@bitg.ch

response following infection of myeloid dendritic cells and monocytes with DNA viruses, and upon infection of a human microglial cell line (CHME3) with single-stranded RNA virus Japanese encephalitis virus (JEV) (Zhang et al, 2013; Manocha et al, 2014), which might point to a cell- and/or virus type-specific function of TRIM21 in the antiviral immune response. Moreover, TRIM21 is the only known intracellular antibody receptor because it binds to virus-bound intracellular antibodies and targets them for proteasomal degradation (Clift et al, 2017).

FAT10 is a ubiquitin-like modifier having two ubiquitin-like domains connected by a flexible linker (Aichem et al, 2018). FAT10 targets its substrates for proteasomal degradation in a ubiquitin-independent manner (Hipp et al, 2005; Aichem et al, 2012; Schmidtke et al, 2014). It is constitutively expressed in organs of the immune system; however, its expression can be induced in almost every cell type upon synergistic stimulation with the pro-inflammatory cytokines TNF and IFNγ (Raasi et al, 1999; Lukasiak et al, 2008). FAT10 can interact with its substrates covalently through its C-terminal diglycine motif as well as in a non-covalent manner (Aichem & Groettrup, 2020). For the covalent conjugation of FAT10 to its conjugation substrates (also called FAT10ylation), a cascade of several enzymes is necessary, as it is also known for ubiquitin and other ubiquitin-like modifiers (Aichem et al, 2010). Briefly, during the process of ubiquitination, ubiquitin binds to the adenylation domain of one of the two known E1 activating enzymes, UBE1 and UBA6 (Ciechanover et al, 1981; Jin et al, 2007; Pelzer et al, 2007), where it becomes adenylated at its C-terminal glycine residue. The activated modifier is then transferred onto the active site cysteine of the same E1 enzyme to form a thioester bond. In the next step, it is transferred to the active site cysteine of a cognate E2 conjugating enzyme by a transthiolation reaction, likewise forming a thioester bond. Finally, different classes of ubiquitin ligases (E3s) catalyze the isopeptide linkage of ubiquitin to the ε-amino-group of an internal lysine residue of a substrate protein (Kerscher et al, 2006; Finley, 2009). UBA6 is so far the only known E1 activating enzyme for FAT10ylation (Chiu et al, 2007). USE1 was the only identified E2 conjugating enzyme of FAT10; however, our recent study has identified additional E2s which can participate in the FAT10ylation of proteins (Aichem et al, 2010; Schnell et al, 2023). Until now, Parkin is the only known E3 ligase for FAT10 which FAT10ylates mitofusin-2 (MFN2) during mitophagy (Roverato et al, 2021).

Besides other functions, FAT10 has been shown to down-regulate the viral-induced type-I IFN response and thus might act as a modulator of the immune response inhibiting excessive cellular damage (Nguyen et al, 2016; Zhang et al, 2016; Mah et al, 2019; Wang et al, 2019; Saxena et al, 2024). Upon infection with lymphocytic choriomeningitis virus (LCMV), FAT10 KO mice secreted higher levels of type-I IFN as compared with the WT mice suggesting that FAT10 inhibits the type-I IFN secretion (Mah et al, 2019). It was further shown that FAT10 expression is induced upon IAV infection in a RIG-I-NFκB signaling pathway-dependent manner (Zhang et al, 2016). RIG-I stability (Nguyen et al, 2016) and RIG-I activation via K63-linked ubiquitination were shown to be regulated by FAT10 (Wang et al, 2019), both causing an inhibition of type-I IFN production. More recently, we have shown that IAV-induced phosphorylation of FAT10 stabilized and increased the activity of OTUB1 which deubiquitinates TRAF3 thereby reducing type-I IFN secretion (Bialas et al, 2019; Saxena et al, 2024).

In this study, we aimed to identify additional targets of FAT10 in the type-I IFN pathway and to this aim we investigated the cross-talk between FAT10 and TRIM21 upon IAV infection. Our data provide strong evidence that TRIM21 gets FAT10ylated and targeted for proteasomal degradation. Coiled-coil and PRYSPRY domains of TRIM21 as well as the C-terminal diglycine motif of FAT10 are important for the TRIM21-FAT10 interaction. Moreover, FAT10 reduces the ubiquitination and thus the activation of TRIM21 and the FAT10-mediated degradation of TRIM21 down-regulates the antiviral type-I IFN response. Overall, our study uncovers an additional mechanism of FAT10-mediated down-regulation of the type-I IFN response upon IAV infection and it points to a highly regulated fine-tuning of the type-I IFN response in the presence of FAT10 to protect tissue damage through prolonged or a deregulated type-I IFN secretion.

## Results

### TRIM21 is covalently modified with FAT10

As we had recently shown that FAT10 down-regulates the viral-induced type-I IFN response by activating the deubiquitinating enzyme OTUB1 (Bialas et al, 2019; Saxena et al, 2024), we were interested to identify additional players within this pathway, which might be regulated by FAT10, as well. Our earlier mass spectrometry screen had identified TRIM21 as a putative substrate of FAT10ylation, suggesting that TRIM21 might be FAT10ylated and targeted for degradation by the 26S proteasome (Aichem et al, 2012). TRIM21 has been shown to be a positive regulator of type-I IFN secretion stabilizing or activating several molecules of the type-I IFN cascade (Li et al, 2023). Thus, we investigated whether FAT10 might regulate TRIM21 stability or its activity. We first investigated if FAT10 directly interacts with TRIM21. For this, HA-tagged FAT10 (from here on referred to as HA-FAT10) and Myc-DDK-tagged TRIM21 (Myc-DDK-TRIM21) were transiently transfected in HEK293T cells, and an immunoprecipitation was performed. HEK293T cells were used for all overexpressing experiments because these cells are easy to transfect and because in these cells, FAT10 expression does not induce apoptosis, as, for example, shown in case of Hela cells (Raasi et al, 2001). Myc-DDK-TRIM21 was immunoprecipitated using an antibody reactive to FLAG (the FLAG tag is also referred to as DDK tag) and subjected to Western blot analysis using an antibody reactive to the HA tag. As seen in Fig 1A, a FAT10-TRIM21 conjugate was detected at ~80 kD, which is in line with the molecular weight predicted for a FAT10-TRIM21 conjugate in our mass spectrometry analysis (Aichem et al, 2012) (Fig 1A, IP: FLAG, IB: HA, lane 4). To confirm the nature of the FAT10-TRIM21 interaction we co-expressed Myc-DDK-TRIM21 and HA-FAT10AV, a conjugation deficient mutant of FAT10 in which the C-terminal diglycine motif was replaced by alanine (A) and valine (V). Immunoprecipitation and Western blot analysis confirmed the covalent nature of the FAT10-TRIM21 interaction because the conjugate was not detected in the presence of HA-FAT10AV (Fig 1B, IP: FLAG, IB: HA, lane 5 versus 6). Of note, Myc-DDK-TRIM21 always appeared as a double band as compared with endogenous TRIM21 as shown in experiments in Figs 4 and 5. We therefore suggest that the second band might represent rather

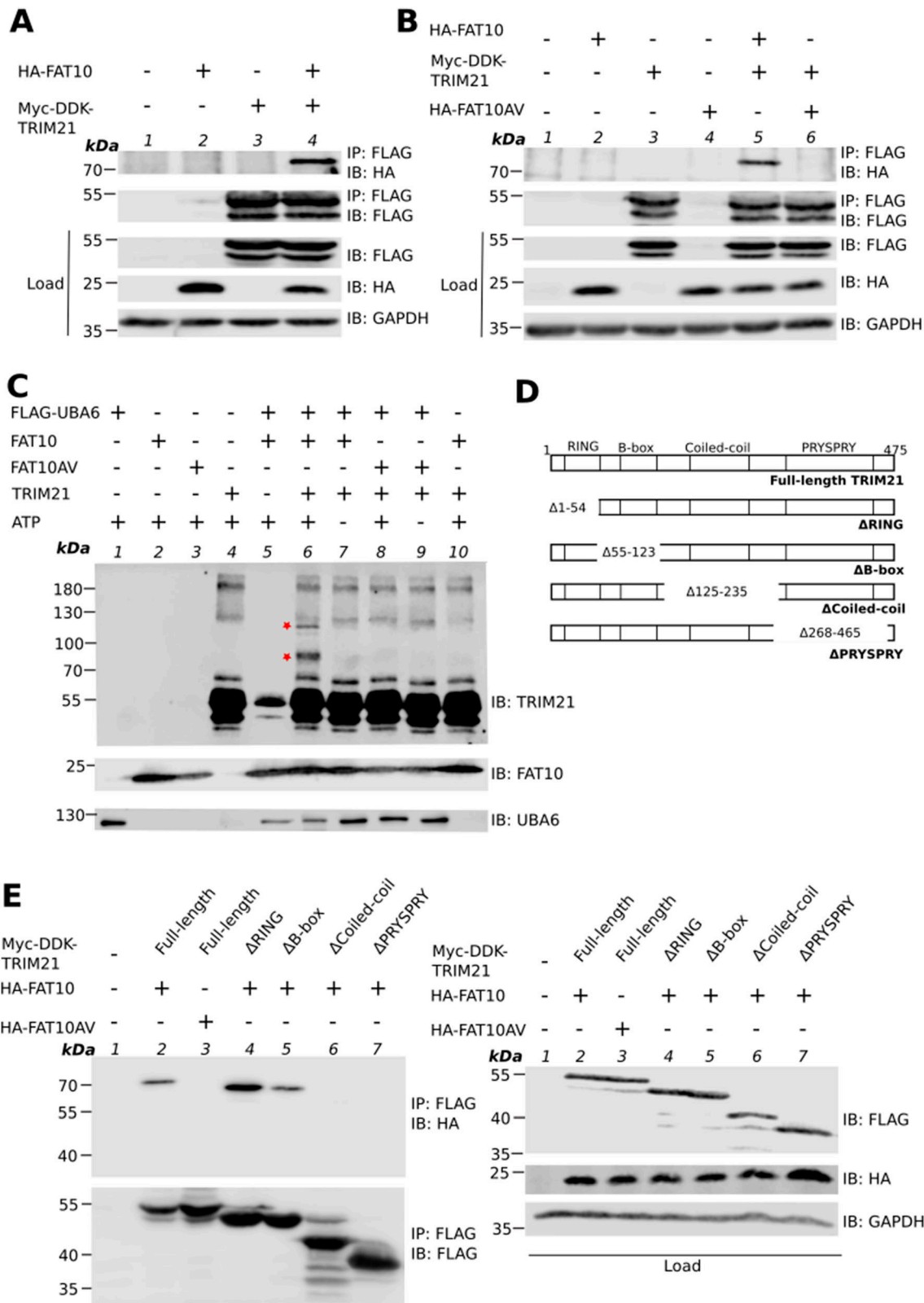

**Figure 1.  TRIM21 is covalently modified with FAT10.**
**(A)** HEK293T cells were transiently transfected with expression constructs for HA-FAT10 and Myc-DDK-TRIM21. After 24 h, cells were harvested and lysed. Cleared lysate was subjected to immunoprecipitation (IP) using FLAG M2 affinity gel, which specifically recognizes the DDK (FLAG) tag. Proteins were visualized by Western blot analysis under reducing conditions (4% 2-ME) using antibodies reactive to HA or FLAG (DDK). GAPDH was used as a loading control. **(B)** HEK293T cells were transiently co-transfected

a posttranslational modification of the Myc-DDK-tag than of TRIM21 itself. Alternatively, it might also represent a specific degradation product of Myc-DDK-TRIM21, which cannot be FAT10ylated anymore.

The covalent conjugation of FAT10 to TRIM21 was further confirmed by an in vitro FAT10ylation assay using recombinant proteins. Our earlier studies have shown that under in vitro conditions, UBA6 alone is sufficient to FAT10ylate many of the FAT10's substrates without the need of an E2 and E3 enzyme (Bialas et al, 2015, 2019; Aichem et al, 2019b; Boehm et al, 2020). Therefore, we used recombinant c-Myc-DDK-TRIM21, FLAG-UBA6, and untagged FAT10 or FAT10AV for the in vitro FAT10ylation assay, in the presence or absence of ATP, as indicated. The TRIM21-FAT10 conjugate was detected in the presence of FLAG-UBA6 and FAT10 in an ATP dependent manner (Fig 1C, IB: TRIM21, lane 6), whereas FAT10 failed to get conjugated to TRIM21 when FAT10AV was used (Fig 1C, IB: TRIM21, lane 8) confirming our finding that the C-terminal diglycine motif of FAT10 is necessary for the FAT10ylation of TRIM21. Interestingly, in our in vitro experiment two FAT10ylation bands for TRIM21 were detected (Fig 1C, lane 6, asterisks), suggesting that under in vitro conditions TRIM21 is modified with FAT10 at more than one lysine residue. This finding is different from our results obtained under overexpressing conditions in HEK293T cells where FAT10 conjugation was detected at a single lysine residue only (Fig 1A). This suggests mechanistic differences between in vitro and in cellulo conditions.

TRIM21 consists of four domains (Oke & Wahren-Herlenius, 2012). Whereas the N-terminal RING, the B-box and the coiled-coil domains are critical for its E3 ligase activity, subcellular localization and dimerization, respectively, the C-terminal PRYSPRY domain is important for its antibody receptor function and interaction with RNA and proteins (Li et al, 2023). To determine which domain of TRIM21 interacts with FAT10, we constructed a series of truncation mutants of TRIM21 by selectively deleting each domain (cartoon in Fig 1D) and transiently co-transfected the truncation mutants together with HA-FAT10 in HEK293T cells. Whereas the RING and the B-box domains were found to be dispensable, deletion of coiled-coil or PRYSPRY domains inhibited TRIM21-FAT10 conjugation, indicating that the coiled-coil and PRYSPRY domains of TRIM21 are critical for its covalent conjugation with FAT10 (Fig 1E, IP: FLAG, IB: HA, lanes 6 and 7).

## UBA6 and USE1 are necessary and sufficient for FAT10ylation of TRIM21

After establishing that TRIM21 is indeed a substrate for FAT10ylation, we aimed to determine the enzymatic cascade catalyzing TRIM21

FAT10ylation. The E1 activating enzyme UBA6 and the E2 conjugating enzyme USE1 are the known E1 and E2 enzymes of the FAT10 conjugation cascade, respectively (Chiu et al, 2007; Aichem et al, 2010). To test whether both enzymes are required for the FAT10ylation of TRIM21, HA-FAT10 and Myc-DDK-TRIM21 were transiently transfected in HEK293 WT, HEK293 UBA6 KO (Aichem et al, 2019b), and HEK293 USE1 KO cells (Aichem et al, 2018). An immunoprecipitation of Myc-DDK-TRIM21 followed by Western blot analysis detected the FAT10-TRIM21 conjugate only in HEK293 WT cells (Fig 2A, lane 5) whereas no conjugate was detected in HEK293 UBA6 KO and USE1 KO cells, indicating UBA6 and USE1 to be necessary for the FAT10ylation of TRIM21. The FAT10-TRIM21 conjugate was again detected in UBA6 KO and USE1 KO cells reconstituted with UBA6 and USE1 expression plasmids, respectively (Fig 2B, lanes 5 and 6), whereas the conjugate was not detected in USE1 KO cells reconstituted with the USE1 active site cysteine mutant USE1-C188A (Fig 2B, lane 7), confirming that the E2 activity of USE1 is important for TRIM21 FAT10ylation. Until recently, USE1 was the only confirmed E2 enzyme for FAT10ylation (Aichem et al, 2010). However, our recent study has identified additional E2 conjugating enzymes for FAT10 conjugation and provided evidence for E2 enzymes which eventually get activated upon TNF treatment (Schnell et al, 2023). To test whether any other E2 conjugating enzyme might FAT10ylate TRIM21 in the absence of USE1, HEK293 WT, UBA6 KO and USE1 KO cells were treated with TNF for 24 h. The FAT10-TRIM21 conjugate was detected in HEK293 WT cells irrespective of TNF treatment (Fig 2C, lanes, 2 and 5). Activation of other E2 enzymes by TNF failed to catalyze the FAT10ylation of TRIM21 in HEK293 UBA6 KO and USE1 KO cells (Fig 2C, lanes 6 and 7), confirming that USE1 is the specific E2 enzyme for the FAT10ylation of TRIM21. Thus, we concluded that UBA6 and USE1 are necessary and sufficient for the FAT10ylation of TRIM21.

## FAT10 targets TRIM21 for 26S proteasomal degradation

Previous studies have shown that FAT10 targets its substrates for degradation by the 26S proteasome (Hipp et al, 2005; Schmidtke et al, 2014). Therefore, it was determined whether the FAT10-TRIM21 conjugate is subjected to degradation by the 26S proteasome, as well. To monitor the stability of the FAT10-TRIM21 conjugate, a cycloheximide (CHX) chase assay was performed in HEK293T cells transiently expressing HA-FAT10 and Myc-DDK-TRIM21. Where indicated, in addition to CHX, cells were also treated for 5 h with the proteasome inhibitor MG132 to inhibit the catalytic activity of the 26S proteasome. Immunoprecipitation followed by Western blot analysis showed a time-dependent degradation of the FAT10-TRIM21 conjugate (Fig 3A, IP: FLAG, IB: HA) which was confirmed

with expression constructs for HA-FAT10, its conjugation incompetent variant HA–FAT10AV and Myc-DDK-TRIM21. After 24 h, cells were harvested and lysed. Cleared lysate was subjected to immunoprecipitation (IP) using FLAG M2 affinity gel. Proteins were visualized by Western blot analysis under reducing conditions (4% 2-ME) using antibodies reactive to HA or FLAG. GAPDH was used as loading control. **(C)** In vitro FAT10ylation assay was performed using recombinant proteins. FLAG–UBA6 was incubated with tagless recombinant FAT10 or FAT10AV and C–Myc-DDK-TRIM21 for 30 min at 37°C. Reaction was stopped by adding 4x sample buffer and boiling. SDS–PAGE and subsequent Western blotting was performed using the indicated antibodies under reducing conditions (4% 2-ME). Asterisks indicate TRIM21–FAT10 conjugates. **(D)** Schematic representation of full-length and truncation mutants of human TRIM21 generated and used in this study. The truncations were constructed using site directed mutagenesis. **(E)** HEK293T cells were transiently transfected with expression plasmids for HA–FAT10 or HA–FAT10AV and full length or truncation mutants of Myc-DDK-tagged TRIM21. After 24 h, cells were collected and lysed. Cleared cell lysate was subjected to immunoprecipitation using FLAG–M2 affinity gel. SDS–PAGE and Western blot analysis was performed under reducing conditions (4% 2-ME) using antibodies reactive to HA or FLAG (DDK). GAPDH was used as loading control. Shown is one representative experiment out of three independent experiments with similar outcomes for all experiments. Source data are available for this figure.

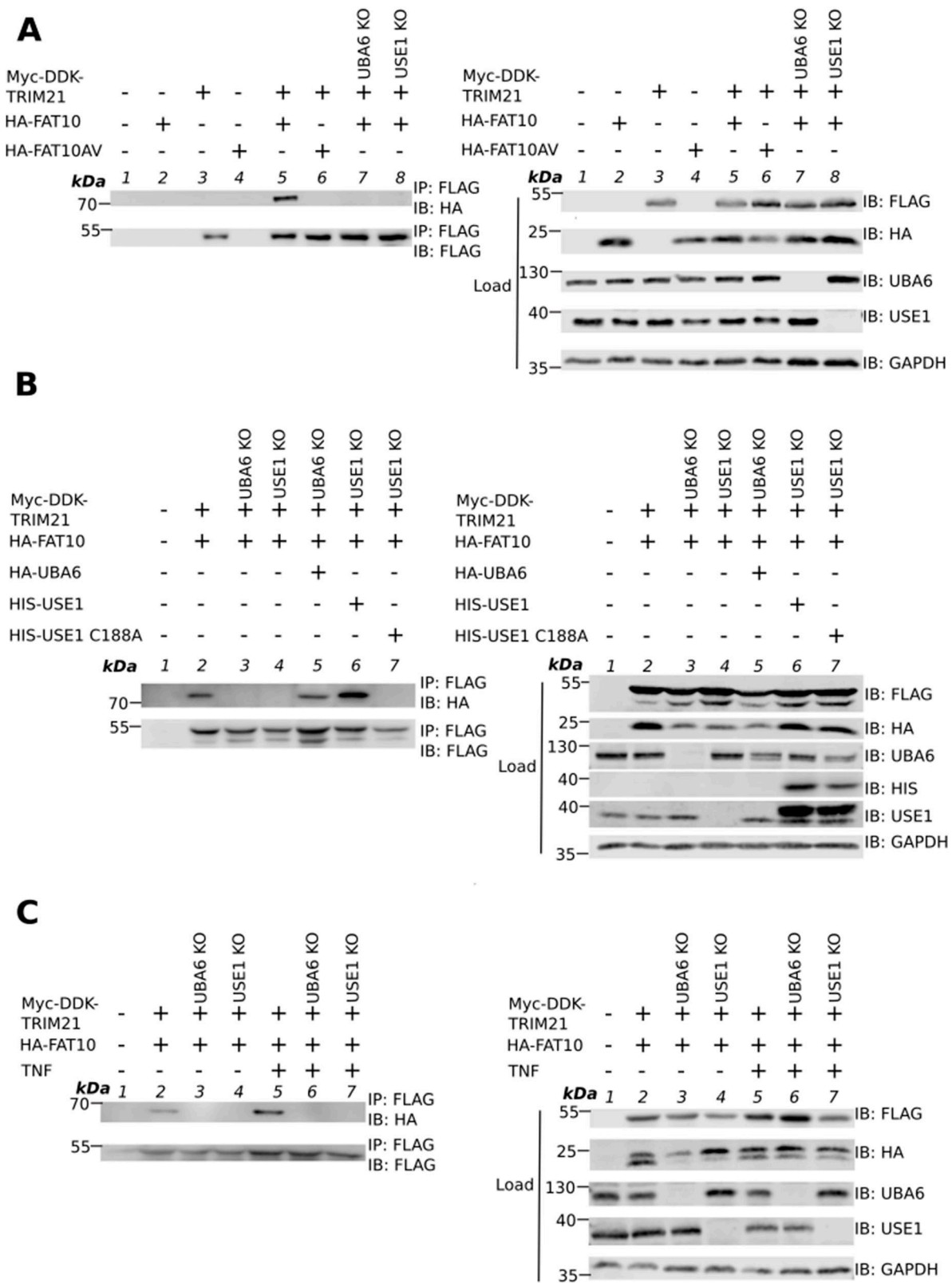

**Figure 2. FAT10ylation of TRIM21 is catalyzed by the E1 UBA6 and the E2 USE1.**

**(A)** HEK293 WT, or where indicated UBA6 KO and USE1 KO cells, were transiently transfected with HA–FAT10 or HA–FAT10AV and Myc–DDK–TRIM21 expression plasmids. After 24 h, cells were collected and lysed. Cleared cell lysate was subjected to SDS–PAGE and subsequent Western blotting under reducing conditions (4% 2-ME) using antibodies reactive to HA or FLAG (DDK). GAPDH was used as loading control. **(B)** HEK293 WT, or UBA6 KO and USE1 KO cells were transiently transfected with expression

by the densitometric quantification of the fluorescence (Licor) and ECL signal intensities (Fig 3B). Of note, the total TRIM21 level was not affected in the presence of FAT10 under these conditions in HEK293T cells (Fig 3A, IP: FLAG, IB: FLAG), indicating that only a small fraction of the total TRIM21 is targeted for FAT10ylation and proteasomal degradation.

## FAT10 reduces TRIM21 levels and its ubiquitination upon IAV infection

In our previous study, we have shown that FAT10 down-regulates IFNβ secretion upon influenza A virus (IAV) infection in lung adenocarcinoma (A549) cells because A549 cells stably expressing FLAG-FAT10 secreted less IFNβ as compared with A549 WT cells infected with IAV (Saxena et al, 2024). In addition, IAV infection was shown to induce the expression of TRIM21 (Yang et al, 2009; Xue et al, 2018; Li et al, 2020). Therefore, we investigated the functional significance of FAT10ylation of TRIM21 in A549 cells infected with IAV. A549 cells were used instead of HEK293T cells because these cells are robustly infected with IAV and are capable to produce and to secrete IFNβ, thus, mimicking in vivo conditions. First, we tested if FAT10 interacts with TRIM21 in these cells, as well. A549 cells were infected with IAV and treated with TNF/IFNγ for 24 h to induce the expression of endogenous FAT10. Immunoprecipitation of TRIM21 using an antibody reactive to TRIM21 followed by Western blotting was performed. In addition to binding to TRIM21 reactive antibodies (Fig S1A, lanes 4–6), TRIM21 was found to bind to IgG control antibodies, as well (Fig S1A, lanes 1–3). In the same cellular system, immunoprecipitation of endogenous FAT10 was performed using a highly specific, monoclonal FAT10-reactive antibody (clone 4F1 [Aichem et al, 2010]) followed by Western blot analysis. TRIM21 was detected in FAT10 immunoprecipitated samples (Fig S1B, lanes 4–6); however, nonspecific binding of TRIM21 was also detected in IgG control samples (Fig S1B, lanes 1–3). In addition, immunoprecipitation using cell lysates from A549 cells stably expressing FLAG-FAT10 and infected with IAV, showed likewise a nonspecific binding of FAT10 in the IgG control of the immunoprecipitation (Fig S1C, lanes 2, 3 versus lanes 5, 6) in addition to the nonspecific binding of TRIM21 to IgG control antibodies (Fig S1C, lanes 1–3). TRIM21 is a strong intracellular antibody receptor which binds to virus-bound antibodies entering inside the cells and targets them for proteasomal degradation through antibody-dependent intracellular neutralization (Mallery et al, 2010; McEwan et al, 2013; Clift et al, 2017; Caddy et al, 2021). As confirmed by our results, due to these antibody-binding properties of TRIM21, a classical immunoprecipitation is not possible to investigate the conjugation of endogenous TRIM21 and FAT10. Therefore, we used an alternative

approach in which Ni-IDA (Nickel-Iminodiacetic acid) beads mediated the pull-down of 6xHIS-3xFLAG-FAT10 in a semi-endogenous system, where A549 cells stably expressed 6xHIS-3xFLAG-FAT10 together with endogenous TRIM21. The plasmid used to generate stable FLAG-FAT10 expressing A549 cells contained a 6xHIS tag in addition to the 3xFLAG tag in its N-terminus (Chiu et al, 2007; Saxena et al, 2024) making it suitable to be used for Ni-IDA pull-down assay. Stable A549 6xHIS-3xFLAG-FAT10 cells were infected with IAV (MOI: 1) for 1 h. 24 h post-infection, cells were harvested, and the precleared lysate was incubated with Ni-IDA beads overnight at 4°C. Proteins were separated on SDS–PAGE under reducing condition (4% 2-ME) followed by Western blot analysis using antibodies reactive to TRIM21 or FLAG. A typical smear of FAT10ylated proteins was observed in A549 FLAG-FAT10 pull-down samples when immunoblotted with an antibody reactive to FLAG (Fig 4A, Ni-IDA-PD, IB: FLAG, lanes 2 and 3), validating an efficient pull-down of FLAG-FAT10 using Ni-IDA agarose beads and representing the bulk of all FAT10ylated proteins having different molecular weights. The TRIM21-FAT10 conjugate was detected at ~80 kD in A549 FLAG-FAT10 cells irrespective of IAV infection (Fig 4A, lanes 2 and 3). Taken together, we could show that TRIM21 is modified with FAT10 under semi-endogenous cellular conditions, meaning by investigating the modification of endogenous TRIM21 with overexpressed FAT10, using Ni-IDA pull-down of 6xHIS-3xFLAG-FAT10 in A549 cells, bypassing the need for the antibody-mediated immunoprecipitation.

Intrigued by our observation of FAT10 mediated targeting of TRIM21 to proteasomal degradation in HEK293T cells, we tested the levels of endogenous TRIM21 in A549 WT and FLAG-FAT10 cells infected with IAV. Cells were infected with IAV (MOI: 1). After 24 h, cells were lysed and total cell lysate was subjected to SDS–PAGE followed by Western blotting with the indicated antibodies (Fig 4B). Whereas there was no difference in TRIM21 protein levels in uninfected A549 WT and FLAG-FAT10 cells (Fig 4B, lanes 1 and 3, and Fig 4C bars 1 and 2), a significant reduction in TRIM21 protein levels was observed in A549 FLAG-FAT10 cells as compared with the WT cells when they were infected with IAV (Fig 4B, lanes 2 and 4, and Fig 4C, bars 3 and 4). This is in contrast to the observation we have made in HEK293T cells overexpressing HA–FAT10 and Myc-DDK-TRIM21 (Fig 3), eventually indicating a stronger FAT10-mediated degradation of TRIM21 upon IAV infection in A549 cells.

TRIM21 is an E3 ligase which gets activated upon viral infection by auto-ubiquitination (Clift et al, 2017). To investigate, if FAT10 might have an impact on the TRIM21 activation, we investigated the auto-ubiquitination of TRIM21 in the presence of FAT10 upon IAV infection. A549 WT and A549 FLAG–FAT10 cells were infected with IAV (MOI: 1). 24 h post-infection, cells were harvested, lysed, and

---

constructs for HA–FAT10 and Myc-DDK-TRIM21. Where indicated, HA-UBA6 expression plasmid was transiently transfected in HEK293 UBA6 KO cells, and HIS–USE1 or HIS-USE1 C188A expression plasmids were transiently transfected in HEK293 USE1 KO cells. After 24 h, cells were collected and lysed. Cleared cell lysates were subjected to SDS–PAGE and subsequent Western blotting was performed under reducing conditions (4% 2-ME) using antibodies reactive to HA or FLAG (DDK). GAPDH was used as loading control. In case of UBA6 KO cells, a 4x times amount of the cell lysate was used for immunoprecipitation because overexpression of proteins is always low in this cell line. **(C)** HEK293 WT, UBA6 KO and USE1 KO cells were transiently transfected with expression constructs for HA–FAT10 and Myc–DDK–TRIM21. Where indicated, cells were treated with 600 U/ml TNF. After 24 h, cells were collected, lysed, and cleared cell lysate was subjected to immunoprecipitation using FLAG M2 affinity gel, which specifically recognizes the DDK (FLAG) tag. SDS–PAGE and Western blotting was performed under reducing conditions (4% 2-ME) using antibodies reactive to HA or FLAG. GAPDH was used as loading control. Shown is one representative experiment out of three independent experiments with similar outcomes for each experiment. Source data are available for this figure.

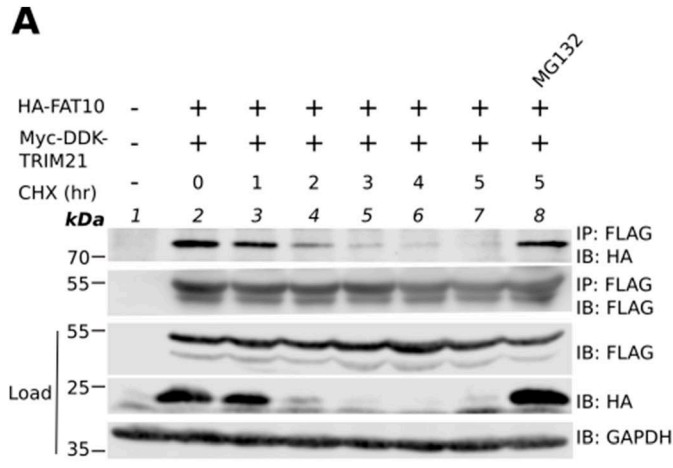

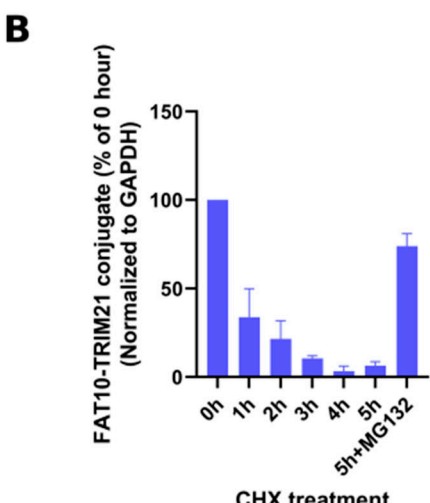

**Figure 3. FAT10 targets TRIM21 for degradation by the 26S proteasome.**
**(A)** HEK293T cells were transiently transfected with expression constructs for HA–FAT10 and Myc–DDK–TRIM21. After 24 h, cells were treated with cycloheximide and/or MG132 for the indicated time points. Cleared cell lysates were subjected to immunoprecipitation using FLAG M2 affinity gel, which is specific for the DDK-tag. SDS–PAGE and Western blot analysis was performed under reducing conditions (4% 2-ME) using antibodies reactive to HA or FLAG (DDK). GAPDH was used as loading control. Shown is one representative experiment out of three independent experiments with similar outcomes. **(B)** Densitometric quantification of the FAT10-TRIM21 conjugate fluorescent signal, normalized to the respective GAPDH fluorescent signal. The value of untreated sample was set to 100%. Shown is the mean three independent experiments with similar outcomes. Source data are available for this figure.

subjected to immunoprecipitation with an antibody reactive to TRIM21. SDS–PAGE followed by Western blotting was performed using antibodies reactive to ubiquitin (Ub) or TRIM21. No visible reduction in the ubiquitination of TRIM21 was observable in A549 WT cells infected with IAV as compared with uninfected WT cells (Fig 4D, lanes 1 and 2). However, we found a strong reduction of TRIM21 ubiquitination in A549 FLAG–FAT10 expressing cells infected with IAV as compared with IAV-infected A549 WT cells (Fig 4D, lane 4 as compared with lane 2), or uninfected A549 FLAG–FAT10 cells (Fig 4D,

lane 4 as compared with lane 3). Of note, equal amounts of immunoprecipitated TRIM21 were observed in lanes 1, 3, and 4 and only slightly, IAV-mediated increased TRIM21 levels were observed in lane 2. Nevertheless, we saw a considerable reduction in ubiquitination of TRIM21 in cells overexpressing FLAG–FAT10 and infected with IAV (Fig 4D, IP: TRIM21, IB: Ub, lane 4 versus lanes 1–3). These data suggest that FAT10 inhibits the bulk ubiquitination of TRIM21 and thus modulates the TRIM21 E3 ligase activity upon IAV infection.

## FAT10-mediated inhibition of TRIM21 contributes to the down-regulation of the antiviral type-I IFN response

Since FAT10 and TRIM21 both function in the regulation of the type-I IFN pathway, we aimed to investigate if FAT10-mediated inhibition of TRIM21 might have an impact on the type-I IFN response upon IAV infection. A549 TRIM21 KO and A549 TRIM21 KO/FLAG–FAT10 cells were generated using the CRISPR/Cas9 technology with guide RNA targeting the human *TRIM21* gene. A549 WT, FLAG–FAT10, TRIM21 KO, and TRIM21 KO/FLAG–FAT10 cells were infected with IAV (MOI: 1). 24 h post-infection, supernatant medium and cell pellets were collected for IFNβ ELISA and Western blot analysis, respectively. As a control for the expression of all involved proteins in the different cell types, cell pellets were lysed and cleared cell lysates were subjected to SDS–PAGE followed by Western blotting with the indicated antibodies (Fig 5A). Again, in A549 cells the endogenous TRIM21 protein level was found to be significantly reduced in the presence of FLAG–FAT10 upon IAV infection (Fig 5A and quantification in Fig S2A). Whereas uninfected cells did not produce detectable amounts of IFNβ (Fig S2B), IFNβ ELISA showed a significant reduction in IFNβ levels in A549 FLAG–FAT10 and TRIM21 KO cells (Fig 5B, bars 2 and 3), as compared with the WT cells (Fig 5B, bar 1) when the cells were infected with IAV, confirming that FAT10 down-regulates, and that TRIM21 activates the antiviral IFNβ response. Moreover, a combinatorial reduction in IFNβ level was observed in A549 TRIM21 KO/FLAG-FAT10 cells infected with IAV (Fig 5B, bar 4) which was significantly less than in IAV infected FLAG-FAT10 or TRIM21 KO cells, suggesting that the reduction in IFNβ expression by FAT10 overexpression and TRIM21 knockout is additive. As a confirmation of the obtained results, IAV titers were determined to ensure that all cell types were infected with the same virus load (Fig S2C). Real-time PCR further confirmed that the observed down-regulation of secreted IFNβ protein correlated with a reduced IFNβ mRNA expression and thus was due to a reduced transcription and not due to a degradation of the IFNβ protein (Fig S2D).

Next, we investigated the effect on IFNβ expression upon overexpressing TRIM21 in the presence of FAT10. The idea was to test whether overexpression of TRIM21 in the presence of FAT10 could rescue the inhibitory effect of FAT10 on IFNβ secretion. A549 cell lines stably expressing mCherry-TRIM21 or mCherry-TRIM21/FLAG-FAT10 were generated using lentiviral transduction of a mCherry-TRIM21 expression plasmid in A549 WT and A549 FLAG-FAT10 cells, respectively. We observed similar expression patterns and cellular distributions of TRIM21 and FAT10 in A549 cells infected with IAV and treated with TNF/IFNγ to induce endogenous FAT10 expression (Fig S3A and B), as well as in A549 cells with stable expression of mCherry-TRIM21/FLAG–FAT10 infected with IAV (Fig S3C). A549 WT,

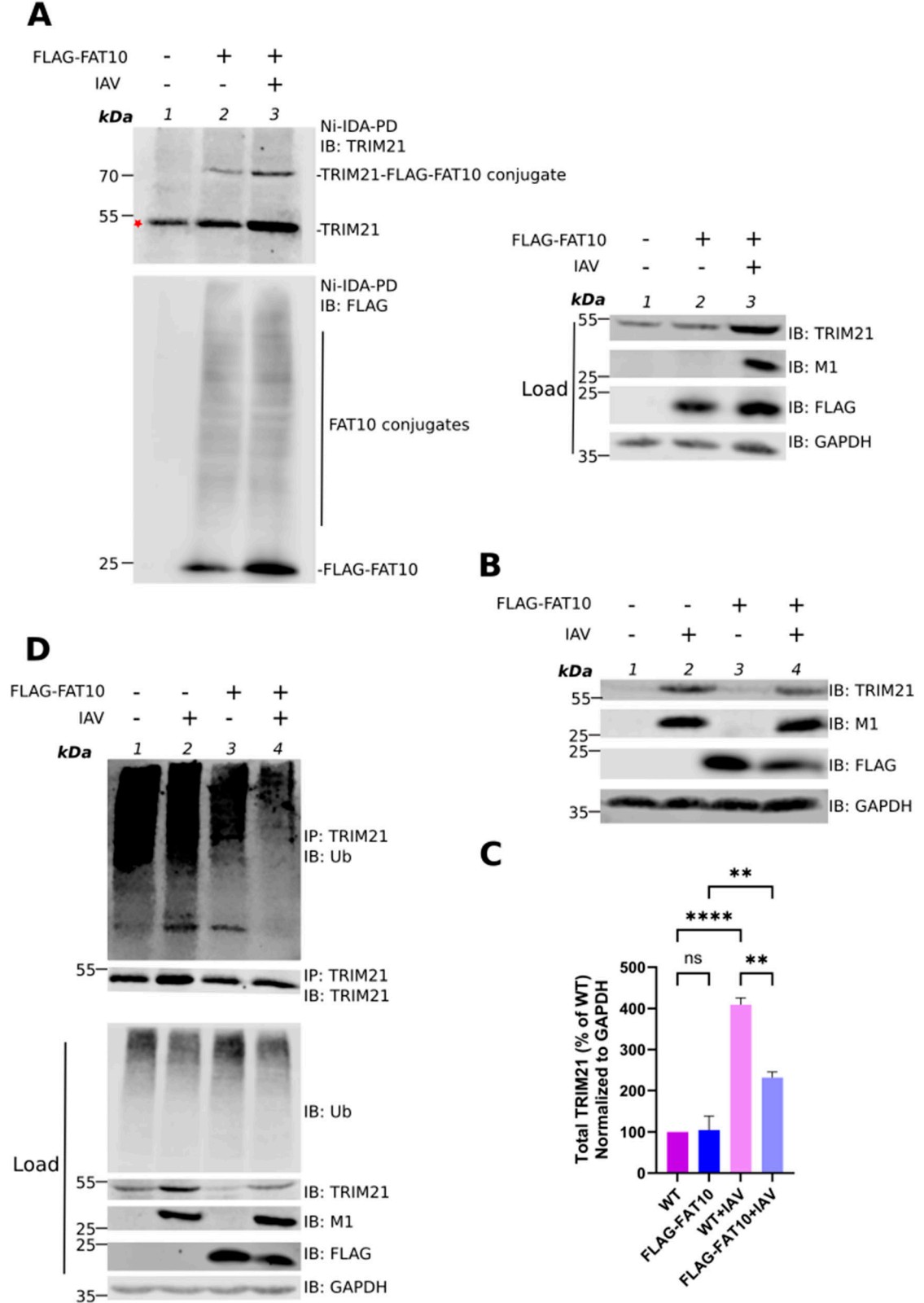

**Figure 4. FAT10 modulates the stability of TRIM21 upon influenza A virus (IAV) infection.**
**(A)** A549 FLAG-FAT10 cells were infected with IAV (MOI: 1) as indicated and uninfected A549 cells were used as control. After 24 h, cells were harvested and lysed. Precleared cell lysates were subjected to Ni–IDA pull-down assay and incubated overnight at 4°C. Beads were washed four times with lysis buffer. SDS–PAGE under reducing conditions (4% 2-ME) followed by Western blotting was performed using the indicated antibodies. GAPDH was used as the loading control. Asterisk marks

FLAG–FAT10, mCherry-TRIM21 and mCherry-TRIM21/FLAG–FAT10 cells were infected with IAV (MOI: 1) as described above. 24 h post infection, supernatant media and cell pellets were collected for IFNβ ELISA and Western blot analysis, respectively. As control for the expression of all relevant proteins, cell pellets were lysed and cleared cell lysates were subjected to SDS–PAGE followed by Western blotting with the indicated antibodies (Fig 5C). As already observed in Figs 5A and S2A, endogenous TRIM21 protein levels were found to be reduced in the presence of FLAG–FAT10 upon IAV infection in A549 cells (Fig 5C, IB TRIM21, lanes 6 and 8 as compared with lanes 5 and 7, respectively and quantification in Fig S2E). Uninfected cells again did not express detectable amounts of IFNβ as measured by ELISA (Fig S2F). Notably, FLAG–FAT10 mediated reduction of IFNβ in A549 FLAG–FAT10 cells (Fig 5D, bar 2) was rescued by the overexpression of mCherry-TRIM21 in mCherry-TRIM21/FLAG–FAT10 cells (Fig 5D, bar 4), indicating that FAT10 mediated reduction of IFNβ is dependent on the stability of TRIM21. Moreover, the increased IFNβ secretion observed in A549 mCherry-TRIM21 cells (Fig 5D, bar 3) was again reduced by FLAG-FAT10 expression in mCherry-TRIM21/FLAG-FAT10 cells (Fig 5D, bar 4). Also under these experimental conditions, viral titers were equal in all infected cells (Fig S2G) and the IFNβ mRNA levels (Fig S2H) correlated with the secreted IFNβ protein amounts shown in Fig 5D, confirming that the observed down-regulation of IFNβ was mediated on the transcriptional level. Taken together, our data from Fig 5B and D suggest that FAT10-mediated down-regulation of IFNβ is in parts mediated through the inhibitory effect of FAT10 on TRIM21.

To furthermore strengthen our data, we induced endogenous FAT10 expression by stimulation of A549 WT, TRIM21 KO, and mCherry-TRIM21 cells with TNF/IFNγ for 24 h. FAT10 KO cells were used as a control. Cells were either infected with IAV (MOI: 0.5) alone, or treated with TNF/IFNγ for 24 h before IAV infection (MOI: 0.5) to induce the expression of FAT10. After 12 h of infection, cell culture supernatants and cell pellets were collected and subjected to IFNβ ELISA. As a control for the expression of all proteins, cell pellets were lysed and cleared cell lysates were subjected to SDS–PAGE and Western blotting using the indicated antibodies (Fig S4A). To detect endogenous FAT10 expression, A549 WT, TRIM21 KO, mCherry-TRIM21, and FAT10 KO cells were treated with TNF/IFNγ for 24 h and an immunoprecipitation was performed using an antibody reactive to FAT10 (clone 4F1). Endogenous FAT10 expression was detected in A549 WT, TRIM21 KO, and mCherry-TRIM21 (Fig S4B lanes 2, 4, and 6) but not in FAT10 KO cells (Fig S4B lane 8), as expected. IFNβ ELISA using the cell culture supernatants showed a significant reduction in IFNβ secretion upon endogenous FAT10 induction in A549 WT and TRIM21 KO cells but not in FAT10 KO cells (Fig 5E). In addition, endogenous FAT10 expression did not affect IFNβ

secretion in A549 mCherry-TRIM21 (Fig 5E) in which TRIM21 levels were high through the overexpression of mCherry-TRIM21, confirming our observation that FAT10 mediated down-regulation of IFNβ is dependent on TRIM21.

FAT10 has been reported to inhibit type-I IFN secretion in C57BL/6 mice upon LCMV infection (Mah et al, 2019). So, next we wanted to investigate if FAT10 modulates the stability of TRIM21 in mice upon LCMV infection. Age- and sex-matched WT C57BL/6 and FAT10 KO C57BL/6 mice were infected with 200 pfu of LCMV-WE intravenously. On day three post-infection, mice were sacrificed and organs were harvested. Real-time quantitative PCR showed induction of FAT10 expression in the liver on day three post-LCMV infection (Fig S5A), similar to its induction in mice splenocytes, as shown earlier by our group (Mah et al, 2019). Cleared cell lysates were prepared from mouse liver and spleen followed by bicinchoninic acid (BCA) assay to estimate the protein content in the samples. 40 μg of protein samples were subjected to SDS–PAGE under reducing conditions (4% 2-ME) followed by Western blotting with antibodies reactive to mouse TRIM21 and GAPDH, which was used as the loading control. No significant difference in TRIM21 levels in the infected and uninfected liver samples (Fig S5B and C) or the spleen samples (Fig S5D and E) was detectable on day three post-LCMV infection. These data suggest that FAT10 does not affect the stability of TRIM21 in mice 3 d post-LCMV infection.

## Discussion

Localization of the *fat10* gene in the MHC gene cluster, its expression in organs of the immune system, and the induction of its expression upon stimulation with pro-inflammatory cytokines strongly implies a FAT10 function in the regulation of the immune response (Lee et al, 2003; Lukasiak et al, 2008). A mass spectrometry analysis of FAT10ylated proteins identified 571 putative FAT10-interacting proteins. In this screen, TRIM21 was identified as a putative FAT10 substrate (Aichem et al, 2012). In the current study, we have now confirmed the result of this mass spectrometry analysis in HEK293T and A549 lung epithelial cells. We show that TRIM21 is covalently modified with FAT10 leading to its degradation by the 26S proteasome, and that the TRIM21-FAT10 conjugate formation is inhibited when the C-terminal diglycine motif of FAT10 is mutated to alanine and valine residues. Moreover, we provide strong evidence that FAT10 inhibits the activation of TRIM21 by interfering with TRIM21 auto-ubiquitination and that both results in a down-regulation of the viral induced type-I IFN response.

TRIM21 belongs to the TRIM family of E3 ligases consisting of conserved N-terminal RING, B-box and coiled-coil domains and

---

nonspecific binding of TRIM21 to the beads. **(B)** A549 WT and FLAG–FAT10 cells were infected with IAV (MOI: 1), as indicated. After 24 h, cells were harvested and lysed. Cleared cell lysates were subjected to SDS–PAGE under reducing conditions (4% 2-ME) followed by Western blotting using the indicated antibodies. GAPDH was used as the loading control. **(C)** Densitometric quantification of the fluorescent signal obtained in (B). The total TRIM21 fluorescent signal was normalized to GAPDH. The signal intensity of other samples was calculated relative to uninfected A549 WT cells in which the normalized TRIM21 signal was set to 100%. **(D)** A549 WT and A549 FLAG-FAT10 cells were infected with IAV (MOI: 1), as indicated. After 24 h, cells were harvested and lysed. Cleared cell lysates were subjected to immunoprecipitation with an antibody reactive to TRIM21. SDS–PAGE under reducing conditions (4% 2-ME) followed by Western blotting was performed using the indicated antibodies. GAPDH was used as loading control. Shown is one representative experiment out of three independent experiments with similar outcomes for each experiment. Error bars in (C) indicate SD (n = 3). *P < 0.05 (one-way anova), ns means not significant.
Source data are available for this figure.

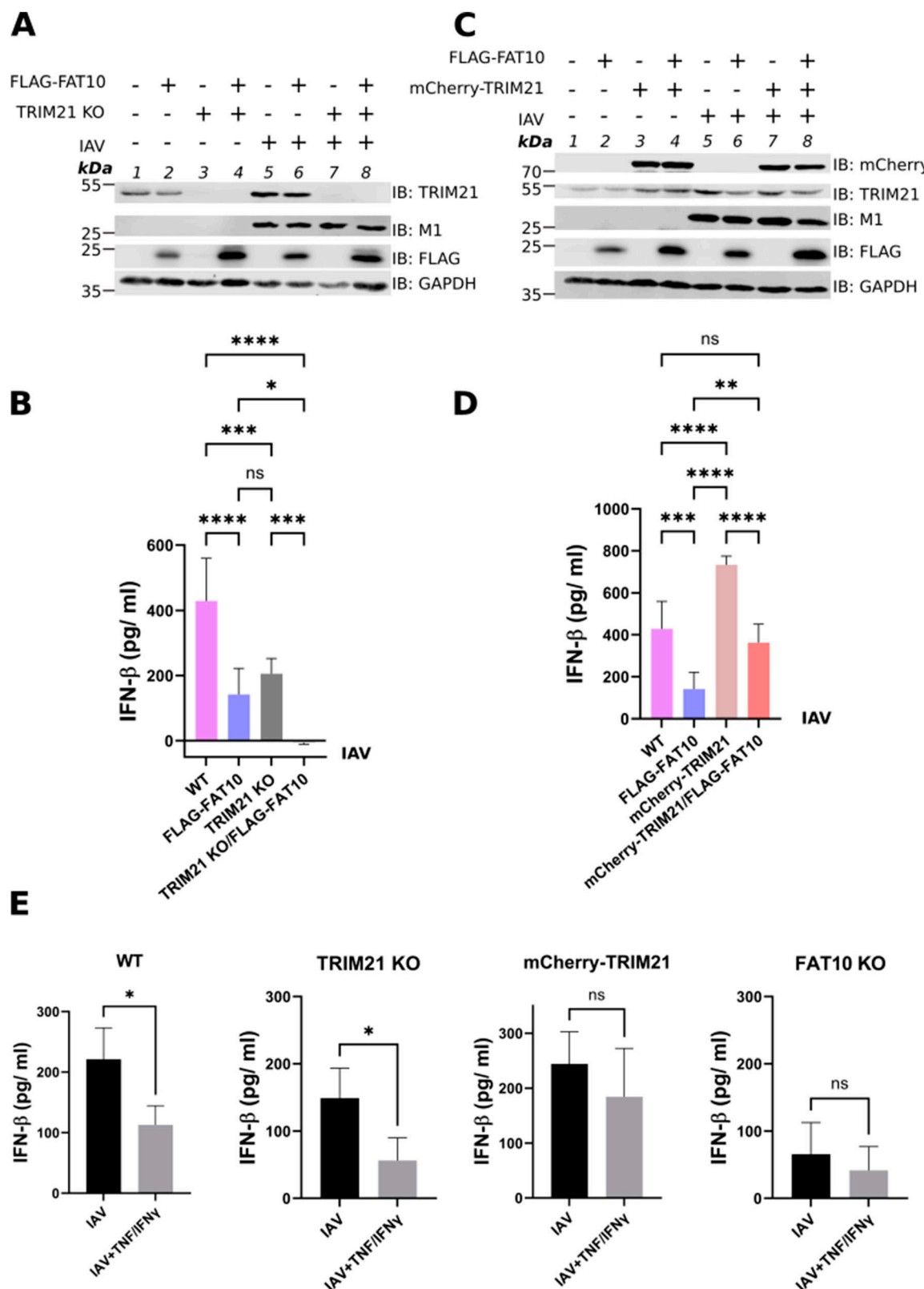

**Figure 5. FAT10-mediated inhibition of TRIM21 down-regulates IFNβ secretion upon influenza A virus (IAV) infection.**
**(A)** A549 WT, FLAG–FAT10, TRIM21 KO, TRIM21 KO/FLAG–FAT10 cells were infected with IAV (MOI: 1). After 24 h, supernatants and cell pellets were collected. Cell pellets were lysed and cleared cell lysates were subject to SDS–PAGE under reducing conditions (4% 2-ME) followed by Western blot analysis with the indicated antibodies. GAPDH was used as the loading control. **(B)** IFNβ ELISA was performed with the cell culture supernatants from (A). **(C)** A549 WT, FLAG–FAT10, mCherry-TRIM21, mCherry-

diverse C-terminal PRYSPRY domains (Oke & Wahren-Herlenius, 2012). We found that the coiled-coil and the PRYSPRY domains of TRIM21 are important for its interaction with FAT10. This is in line with other studies which showed that the PRYSPRY domain of TRIM21 is involved in protein-protein interactions (van Gent et al, 2018; Holwek et al, 2023). Similarly, the coiled-coil domain of TRIM21 is important for the formation of a catalytically active homodimer of TRIM21 (van Gent et al, 2018). In our study, when the coiled-coil domain was truncated, TRIM21 did not get FAT10ylated. This observation indicates that either FAT10ylation takes place at a lysine residue within this region or that the dimerization of TRIM21 might be important for its FAT10ylation, or both. These data are suggestive that a single FAT10 molecule binds to each of the PRYSPRY domain of the dimer, which is then consistent with the molecular weight obtained for the TRIM21–FAT10 conjugate in the mass-spectrometry analysis as well as in the cellular system (Fig 1) (Aichem et al, 2012). Moreover, several members of the TRIM family are involved in the regulation of the type-I IFN signaling pathway and they all share the coiled-coil domain but have different PRYSPRY domains. It will be interesting to investigate if FAT10-mediated degradation is specific to TRIM21 or if FAT10 can in addition regulate the stability of other TRIM proteins sharing structural similarities.

FAT10ylation conjugates single FAT10 molecule to lysine residue of the substrates. Unlike ubiquitin, polymeric chains of FAT10 have so far not been detected, whereas the conjugation of several single FAT10 molecules to the same substrates have been observed, for example, in case of the autophagy receptor p62/SQSTM1 (Aichem et al, 2012) or the FAT10 E3 ligase Parkin (Roverato et al, 2021). We detected two FAT10ylation bands for TRIM21 in our in vitro FAT10ylation experiments (Fig 1C), indicating that TRIM21 can get FAT10ylated at two different lysine residues simultaneously with two FAT10 molecules. Multiple FAT10ylation at different lysine residues under distinct stimuli might have functional consequences such as differential protein interactions, or alterations in ubiquitination. However, further studies are needed to address this hypothesis for TRIM21. Interestingly, in cellulo, we detected only a single FAT10ylation of TRIM21, which might eventually be explained by the fact that in most of our FAT10ylation experiments performed under in vitro conditions with the usage of relatively high protein concentrations, the E1 enzyme UBA6 was sufficient to mediate FAT10ylation of the respective substrates (Bialas et al, 2015, 2019; Aichem et al, 2019b; Boehm et al, 2020). It remains to be investigated, if multiple FAT10ylation of TRIM21 might also occur in cells under certain conditions.

FAT10ylation of TRIM21 was dependent on UBA6 and USE1, the known E1 activating and E2 conjugating enzymes of FAT10, respectively. Our recent study has identified several additional E2 enzymes that can FAT10ylate proteins independent of USE1 and TNF seems to be required for the stimulation or activation of at least

one more, not yet identified FAT10-specific E2 enzyme (Schnell et al, 2023). However, in our study, we did not detect a TRIM21-FAT10 conjugate when we treated USE1 KO cells with TNF suggesting that USE1 is the only and thus specific E2 enzyme for the FAT10ylation of TRIM21 and that no other E2 conjugating enzyme functions redundantly with USE1 in TRIM21 FAT10ylation.

FAT10ylated substrates are targeted for proteasomal degradation (Hipp et al, 2005; Schmidtke et al, 2014) and therefore, we tested if FAT10 targets TRIM21 for degradation, as well. In HEK293T cells, the TRIM21-FAT10 conjugate degraded in a time-dependent manner which was rescued by inhibiting proteasome activity with MG132. However, the stable steady-state level of TRIM21 in the presence of FAT10 indicated that only a minor fraction of TRIM21 is targeted for FAT10-mediated proteasomal degradation under unstimulated and overexpressing conditions (Fig 3). In contrast to this observation in HEK293T cells, we saw that this was not valid for endogenous TRIM21 levels in A549 cells. When A549 cells were infected with IAV, the TRIM21 level was significantly reduced in the presence of FAT10 (Figs 4B and C and S2A and E). This observation suggests that the inhibitory function of FAT10 might eventually further be dependent on the activation of certain adaptor proteins and enzymes involved in the type-I IFN pathway, induced upon IAV infection in A549 cells. Because our experiments in HEK293T cells were performed in the absence of a viral IAV infection under overexpressed conditions, the reduction of TRIM21 steady-state protein amounts might most probably not have been detectable.

TRIM21 is an E3 ligase which auto-ubiquitinates itself and by this gets activated (Fletcher et al, 2015). TRIM21 has been reported to become ubiquitinated by both, K63- and K48-linked chains and K63-linked polyubiquitination is important for its activation (McEwan et al, 2013; Clift et al, 2017). We found that the overall ubiquitination of TRIM21 was reduced in the presence of FLAG-FAT10 when the cells were infected with IAV (Fig 4D). This shows that FAT10 is able to dampen TRIM21 activation not only by inducing its proteasomal degradation but also by inhibiting its activation by ubiquitination. So far, we did not determine which ubiquitin linkage type is affected by FAT10 co-expression; however, this will be subject of future investigations. Despite the covalent modification of proteins with FAT10 and the resulting degradation of the target proteins by the 26S proteasome, we have observed for many FAT10 interaction partners, that a non-covalent interaction with FAT10 altered the activity of the respective target protein. For example, the deubiquitinating enzyme OTUB1 was activated by a non-covalent interaction with FAT10, and the retina-specific phosphodiesterase PDE6 was inhibited by a non-covalent FAT10 interaction. Moreover, both proteins were at the same time also covalent conjugation substrates of FAT10, which were targeted to proteasomal degradation (Bialas et al, 2019; Boehm et al, 2020). In case of TRIM21, we were not able to detect a non-covalent interaction of TRIM21 and

---

TRIM21/FLAG–FAT10 cells were infected with IAV (MOI: 1). After 24 h, supernatants and cell pellets were collected. Cell pellets were lysed and cleared cell lysate was subjected to SDS–PAGE under reducing conditions (4% 2-ME) followed by Western blot analysis with the indicated antibodies. GAPDH was used as the loading control. **(D)** IFN$\beta$ ELISA was performed with the cell culture supernatants from (C). **(E)** A549 WT, TRIM21 KO, mCherry–TRIM21, and FAT10 KO cells were treated with TNF/IFN$\gamma$ for 24 h followed by IAV infection (MOI: 0.5). After 12 h, supernatant and cell pellets were collected. IFN$\beta$ ELISA was performed with the supernatants. Shown is one representative experiment out of at least three independent experiments with similar outcomes for each experiment. Error bars in (B, D, E) indicate SD (n = 3), *$P < 0.05$ (One-way Anova (B, D), student's $t$ test (E)), ns means not significant.
Source data are available for this figure.

FAT10 due to technical reasons. Because TRIM21 is known as intracellular antibody binding protein (Keeble et al, 2008), we were not able to perform standard immunoprecipitations under endogenous conditions, where a putative non-covalent interaction between TRIM21 and FAT10 might be visible. However, due to this fact, we cannot exclude that the inhibitory effects FAT10 exerts onto TRIM21 might be at least in part mediated by a non-covalent interaction of the two proteins, or maybe also by the action of both, covalent modification and non-covalent interaction. Moreover, the interaction of TRIM21 with FAT10 might alter the three-dimensional structure of TRIM21 such that its ubiquitination sites get inaccessible. Alternatively, the TRIM21–FAT10 interaction might inhibit the binding of ubiquitin-specific E2 conjugating enzymes thus inhibiting its ubiquitination. Another possibility would be that FAT10ylation takes place at ubiquitination sites, likewise inhibiting ubiquitination. Additional research will be necessary to clarify the exact mechanism of how FAT10 interferes with TRIM21 ubiquitination.

To investigate if the FAT10 mediated reduction in type-I IFN production is dependent on its inhibitory effect onto TRIM21, we tested the effect of TRIM21 KO in FLAG–FAT10 expressing A549 cells infected with IAV. The idea was that if the negative effect of FAT10 on IFNβ secretion would be completely dependent on TRIM21 we should not see a further reduction in IFNβ levels in TRIM21KO/FLAG-FAT10 cells. Interestingly, we detected an increased, additive reduction of IFNβ secretion in TRIM21 KO cells overexpressing FLAG–FAT10 (Fig 5B). If the inhibitory effect of FAT10 was only mediated by TRIM21 we would have expected to see an increase in IFNβ levels in TRIM21 KO cells overexpressing FLAG–FAT10 which would be at least similar to TRIM21 KO cells. However, an additive effect of FAT10 expression in TRIM21 KO on IFNβ levels indicates

an alternative regulatory mechanism mediated by FAT10, which compensates the production of IFNβ in the absence of TRIM21. We have recently published that the non-covalent interaction of phosphorylated FAT10 with OTUB1 reduced the type-I IFN response by inhibiting TRAF3 poly-ubiquitination (Saxena et al, 2024). IAV infection triggers RIG-I mediated type-I IFN response and TRIM21 expression. To modulate type-I IFN response FAT10 can operate at two levels: targeting TRIM21 for proteasomal degradation, thereby inhibiting TRIM21-mediated activation of MAVS and IRF3 (Yang et al, 2009; Xue et al, 2018), and activating OTUB1 which inhibits TRAF3 activation (Bialas et al, 2019). In TRIM21 KO cells, OTUB1-mediated inhibition of TRAF3 is still mediated by FAT10, which might become strong due to a lack of TRIM21-mediated regulation. This could explain why we see an additive effect of FAT10 expression in TRIM21 KO/FLAG–FAT10 cells. The investigation of FAT10 expression in TRIM21/OTUB1 double knock-out cells will be necessary to further confirm that FAT10-mediated regulation of TRIM21 and OTUB1 results in combined reduction of IFNβ secretion, however, because of pleiotropic functions of TRIM21 and OTUB1 in type-I IFN signaling pathway, it will be difficult to conclude. Moreover, IFNβ reduction seen in FAT10-expressing cells was rescued by overexpressing TRIM21 (Fig 5D), suggesting that the stability of TRIM21 is important for virus-induced IFNβ production. In summary, from these experiments, we conclude that the FAT10-mediated degradation of TRIM21 results into FAT10-mediated down-regulation of antiviral type-I IFN secretion, though other pathways modulated by FAT10 (OTUB1/TRAF3 axis) also contribute to this down-regulation.

Our earlier study had shown that FAT10 KO mice infected with LCMV secreted higher levels of type-I IFNs (Mah et al, 2019). We investigated the stability of TRIM21 in the liver and spleen of WT and FAT10 knockout mice infected with LCMV on day three post-

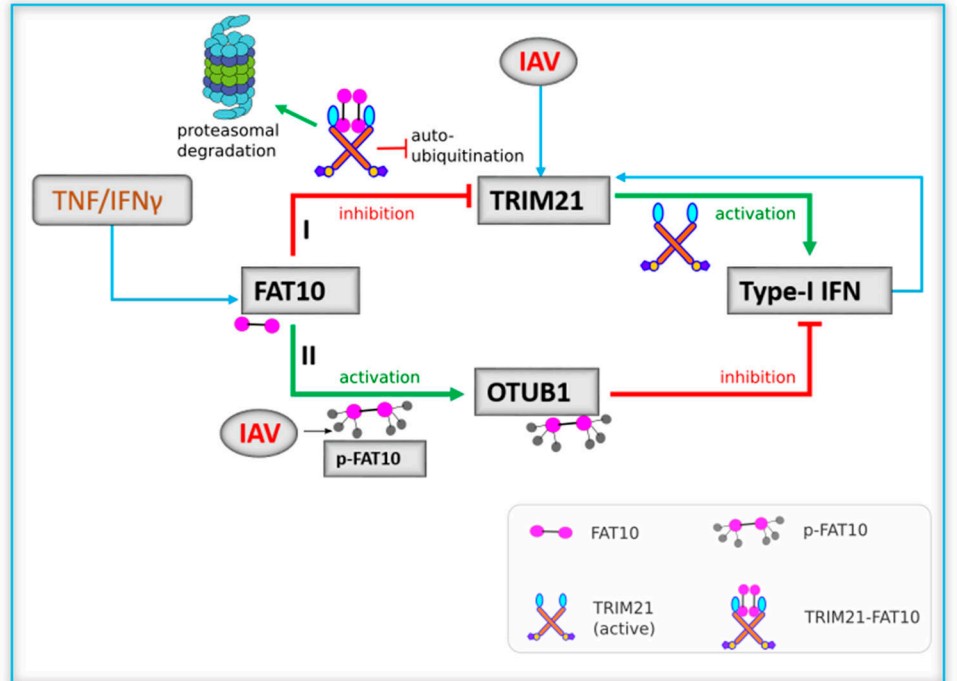

**Figure 6. Cartoon summarizing FAT10 mediated inhibition of type-I IFN signaling.**
TNF/IFNγ induces the expression of FAT10 (blue arrow). FAT10-mediated down-regulation of type-I IFN happens in two ways: (I) influenza A virus infection upregulates TRIM21 expression (blue arrow). TRIM21 positively regulates the antiviral type-I IFN production through a positive feedback loop (blue arrow). FAT10 inhibits TRIM21 by directly binding to its PRYSPRY domain, causing either degradation of TRIM21 by the 26S proteasome, and/or inhibiting TRIM21 auto-ubiquitination, thus down-regulating the production of type-I IFN. (II) FAT10 gets phosphorylated upon influenza A virus infection. Phosphorylated FAT10 stabilizes and activates OTUB1, which inhibits type-I IFN production (Saxena et al, 2024).

infection, when also FAT10 expression was detectable (Fig S5 and [Mah et al, 2019]). However, we did not see a difference in the protein levels of TRIM21 in the liver and the spleen samples of infected FAT10 KO mice as compared with infected WT mice. Therefore, it might be reasonable to investigate TRIM21 protein levels at different time points post-infection. The innate immune response is activated against LCMV infection during the initial days of infection and around day five the adaptive immune response starts (Zhou et al, 2012). Thus, TRIM21 levels in mouse liver and spleen might be measured also on day five and eight post-LCMV infection when the innate immune response is resolving to confirm our data obtained in A549 cells also under in vivo conditions.

In conclusion, we report that TRIM21 is a new FAT10 conjugation substrate and that the TRIM21-FAT10 conjugate is targeted for proteasomal degradation. The coiled-coil and PRYSPRY domains of TRIM21 and the C-terminal diglycine motif of FAT10 are important for TRIM21-FAT10 covalent conjugation in which UBA6 and USE1 function as the E1 activating and E2 conjugating enzymes, respectively. Of note, we shed light on the FAT10-mediated post-translational regulation of TRIM21 modulating its stability. Furthermore, we have shown that upon IAV infection, FAT10 targets TRIM21 for degradation, alters its bulk ubiquitination and diminishes TRIM21 activation. Taken together, we report that the FAT10-mediated down-regulation of the antiviral type-I IFN response is not only mediated by the recently published activation of OTUB1 (Saxena et al, 2024) but also mediated through TRIM21 providing evidence that FAT10 fine-tunes the antiviral type-IFN response by modulating different regulators of the pathway to timely resolve it (Fig 6).

# Materials and Methods

### Cell culture and cell lines

HEK293 WT, HEK293 UBA6 KO cells (Aichem et al, 2018; 2019a; 2019b), HEK293 USE1 KO cells (Aichem et al, 2018; 2019a; 2019b), and MDCK II were cultivated in IMDM (Gibco/Thermo Fisher Scientific), supplemented with 10% FCS (Gibco/Thermo Fisher Scientific), and 1% penicillin/streptomycin (100x) (Gibco/Thermo Fisher Scientific). HEK293T, A549 WT, A549 FLAG-FAT10, A549 mCherry-TRIM21, A549 mCherry-TRIM21/FLAG-FAT10, A549 TRIM21 KO, A549 TRIM21 KO/ FLAG-FAT10, and A549 FAT10 KO were cultivated in DMEM (Gibco/ Thermo Fisher Scientific) supplemented with 10% FCS (Gibco/ Thermo Fisher Scientific) and 1% penicillin/streptomycin (100x) (from Gibco/Thermo Fisher Scientific). All basic media contained GlutaMAX. Cells were cultured in a humid chamber at 37°C and 5% $CO_2$.

### Mice

C57BL/6 mice (H-2b) were originally purchased from Charles River Laboratories, Göttingen, Germany. FAT10-deficient (B6.129S6-Ubdtm1Can) mice were kindly provided by A. Canaan and S.M. Weissman (Yale University School of Medicine, New Haven, USA) (Canaan et al, 2014). FAT10-deficient mice were backcrossed onto

C57BL/6 background for at least 10 generations. 8–10 wk old mice were used for the experiments. Animal experiments were approved by the Review Board of Governmental Presidium Freiburg of the State of Baden-Württemberg, Germany (G-24-008 and G-18/72).

### DNA constructs

Plasmids used for the transient transfection of HEK293 cells were pCMV6-Myc-DDK-TRIM21 (RC202088; Origene), pcDNA3.1-HA-FAT10 (Hipp et al, 2004), pcDNA3.1-HA-FAT10AV (Aichem et al, 2010), pcDNA3.1-HA-UBA6 (Aichem et al, 2010), pcDNA-3.1-His/-A-USE1, and its active site cysteine mutant pcDNA-3.1-His/-A-USE1-C188A (Aichem et al, 2010), lentiviral envelop plasmid pMD2.G (#12259; Addgene), and lentiviral packaging plasmid psPAX2 (#12260; Addgene). pSMPP-mCherry-hTRIM21 was a gift from Leo James (plasmid #104972 [Clift et al, 2017]; Addgene), and pSpCAS9 wt (BB)-2A-GFP targeting human TRIM21 was a gift from Gaudenz Danuser (plasmid # 138295 [Park et al, 2020]; Addgene). pCMV6-Myc-DDK-TRIM21 (RC202088; Origene) was used as template to construct the truncation mutations of TRIM21 by site-directed mutagenesis. The following primer pairs were used as forward primer and reverse primer, respectively, to generate the respective mutations: ΔRING (del aa 1–54) 5′-CGGCAGCGCTTTCTGC-3′ and 5′-GGCGATCGCGGCGG-3′, ΔB-box (del aa 55–123) 5′-GCCATGGTCCCTCTTG-3′ and 5′-GCA-CACAGGACAGAC-3′, ΔCoiled-coil (del aa 125–235) 5′-TGCCA-CAGCTCAGCAC-3′and 5′-GGCGTGGTCACGGTG-3′, ΔPRYSPRY (del aa 268–465) 5′-AATATTGGATCACAAGGATC-3′, and 5′-TGGAGAGGTAA-TATCCAG-3′. All constructs were verified by sequencing (Microsynth AG).

### IAV and infection of cells

The IAV strain A/Regensburg/D6/09 (H1N1pdm09, RB1) was a kind gift of Oliver Planz, University of Tuebingen, Germany, and was produced as described earlier (Mah et al, 2019). The infection was performed as previously described (Saxena et al, 2024). Briefly, 1 × $10^6$ cells in a six well dish or 2.5 × $10^6$ cells in a 100 mm$^2$ dish were seeded. On the next day, the media was discarded followed by 1x washing step with PBS. Cells were incubated with IAV (MOI: 1) in serum-free DMEM medium. After 1 h, the infection medium was replaced with complete DMEM medium. After 24 h of infection, supernatant and cell pellet were collected for IFN$\beta$ ELISA and Western blot analysis, respectively.

### IAV plaque assay

Titers of IAV in the cell culture supernatants of the infected cells were determined on MDCK II cells, as previously described (Baer & Kehn-Hall, 2014). Briefly, 1 × $10^6$ MDCK II cells were seeded in a 12 well dish. On the next day, cells were incubated for 1 h with 300 $\mu$l of 100-fold prediluted and 3-fold serially diluted supernatant cell culture media of the infected cells at 37°C and 5% $CO_2$. Supernatant media from the uninfected control cells were used as the negative control. After the incubation, supernatant media was replaced with 1.5 ml per well overlay media (1:1 ratio of MEM [Gibco/Thermo Fisher Scientific] and 2.5% Avicel [FMC BioPolymer]) and incubated for 48 h at 37°C and 5% $CO_2$. After the incubation, overlay media was

discarded; cells were washed with PBS and fixed by adding 500 $\mu$l of 4% ROTIHistofix (Carl Roth GmbH) for 30 min at 4°C. Cells were washed with PBS and stained with 1% (wt/vol) crystal violet (150 $\mu$l per well) for 15 min at RT in a rocker. Plaques were counted and IAV titer was determined by calculating the plaque forming units (pfu/ml) using the formula:

pfu/ml = Average number of plaques/Dilution*Volume of supernatant added to the well.

### LCMV infection of mice

LCMV-WE was originally obtained from F. Lehmann-Grube (Heinrich Pette Institut, Universität Hamburg, Hamburg, Germany) and propagated on the fibroblast line L929. Age- and sex-matched C57BL/6 mice or FAT10-deficient mice were infected intravenously (i.v.) with 200 pfu of LCMV-WE.

### Transient transfection

Transient expression of plasmids in HEK293T cells was performed by transfection using PEI transfection reagent (408727; Merck). Briefly, when cells reached 60–80% confluency, 8–12 $\mu$g of plasmid DNA (100 mm$^2$ dish) was mixed with the PEI reagent (1:3 DNA to PEI ratio) in "empty" DMEM or IMDM medium without FCS and antibiotics. The empty medium was replaced with complete medium after 5 h of transfection. Cells were incubated at 37°C and 5% CO$_2$ for at least 24 h before harvesting. When required, empty pcDNA3.1-His/-A (#V38520; Invitrogen) was used to balance plasmid amounts. For transfection of A549 cells with CRISPR plasmids, FuGENE transfection reagent (#E2311; Promega) was used at a 1:3 ratio ($\mu$g DNA: $\mu$l FuGENE).

### CHX chase assay

Transfected cells were treated for the indicated time periods with CHX (50 $\mu$g/ml final concentration; Sigma-Aldrich) before cell lysis. Where indicated, MG132 (10 $\mu$M final concentration; Merck) was simultaneously added for 5 h before harvesting to inhibit the proteasome.

### Cell lysates, immunoprecipitation, SDS PAGE and immunoblotting

Cells were harvested and lysed as described earlier (Saxena et al, 2024). Briefly, cell culture medium was discarded and cells were washed 1x with PBS followed by trypsinization at 37°C for 5 min. Complete DMEM was added to neutralize the trypsin, followed by collection of cell suspension. Cells were pelleted by centrifugation at 300$g$ for 5 min. Supernatant was discarded and cells were lysed in Triton-X 100 lysis buffer 20 mM Tris–HCL pH 7.6, 50 mM NaCl, 10 mM MgCl$_2$, 1% Triton X-100, 1x protease inhibitor mix (Mini; cOmplete, EDTA-free Protease Inhibitor Cocktail Tablets [1 tablet per 50 ml; Roche]). Lysis was performed at 4°C for 20 min with intermittent vortexing. Cleared cell lysate was obtained by centrifuging the lysed cells at 16,000$g$ at 4°C for 20 min. Immunoprecipitation was performed as described earlier (Aichem et al, 2019b) by overnight incubation of cleared cell lysate at 4°C with 25 $\mu$l of FLAG-M2 affinity gel (A2220; Merck), or 25 $\mu$l EZview Red

Protein A affinity gel (P6486; Merck) and rabbit anti-TRIM21 antibody (1 $\mu$g) (ab91423, 1:2,000; Abcam), or 25 $\mu$l EZview Red Protein A affinity gel (E3403; Merck) and mouse anti-FAT10 4F1 antibody (1 $\mu$g) (Enzo Life Science [Aichem et al, 2010]). Whole cell lysate was used as the load control. Beads were washed with buffer NET-TN followed by washing with buffer NET-T (1x each) (Aichem et al, 2018). 25 $\mu$l of 4x SDS sample loading buffer supplemented with 4% 2-ME was added and samples were boiled at 95°C for 5 min. Proteins were separated on 10% or 12% SDS acrylamide gels, followed by transfer to 0.45 micron nitrocellulose membrane (Sigma-Aldrich) (as described earlier [Roverato et al, 2021]). Immunoblotting was performed using the following antibodies: mouse anti-FLAG (F1804, 1:3,000; Merck), mouse anti-FLAG (HRP) (A8592, 1:3,000; Merck), rabbit anti-FLAG (F7425, 1:750; Merck), mouse anti-HA (H3663, 1:5,000; Merck), rabbit anti-HA (H608, 1:1,000; Merck), rabbit anti-GAPDH (G9545, 1:10,000; Merck), mouse anti-FAT10 4F1 (1:1,000; Enzo Life Sciences), rabbit anti-FAT10 (1:500; Enzo Life Sciences [Hipp et al, 2004]), mouse anti-M1 (ab22395, 1:1,000; Abcam), rabbit anti-TRIM21 (ab91423, 1:2,000; Abcam), rabbit anti-TRIM21 (ab207728, 1:200; Abcam) ,mouse anti-TRIM21 (sc-25351, 1:500; Santa Cruz), mouse anti-ubiquitin FK2 (1:1,000; Enzo Life Sciences), rabbit anti-UBA6 (1:1,000 [Aichem et al, 2010]; Enzo Life Sciences), rabbit anti-USE1 (1:1,000 [Aichem et al, 2010]; Enzo Life Sciences), rabbit anti-mCherry (43590, 1:1,000; Cell Signaling), HIS (HRP) (A7058-1VL1:1,000; Sigma-Aldrich), mouse anti-M1 antibody (ab22395, 1:1,000; Abcam), 800CW goat anti-mouse IgG (926-332210 1:10,000; Licor), 680RD goat anti-mouse IgG (926-68070 1:10,000; Licor), 800CW goat anti-rabbit antibody (926-32211, 1:10,000; Licor), and 680RD goat anti-rabbit antibody (926-68071, 1:10,000; Licor). Western blots were imaged using Odyssey M Imager (LI-COR Biosciences). For quantification, fluorescent or ECL signals were analyzed with densitometry calculations (Image Studio Software, LI-COR Biosciences), and the values were normalized to the respective proteins in the lysate or to the loading control GAPDH. Cleared cell lysate was prepared from mouse spleen and liver samples infected with LCMV as described above using RIPA lysis buffer (50 mM Tris-buffered HCl, pH 7.5,150 mM NaCl, 1% NP-40, 0.5% SDS). Protein content was determined by bicinchoninic acid (BCA) assay (Thermo Fisher Scientific). 30–40 $\mu$g protein was used to perform SDS–PAGE and Western blot analysis as described above.

### Ni-IDA pull-down assay

Cells were harvested and lysed as described above. Precleared cell lysate was incubated overnight with 5 $\mu$g of protino Nickel-iminodiacetic acid beads (Ni-IDA) (#745210.120; Macherey-Nagel) at 4°C with rotation. After the incubation, beads were washed four times with the lysis buffer (as described above). 25 $\mu$l of 4x SDS sample loading buffer supplemented with 4% 2-ME was added and samples were boiled at 95°C for 5 min.

### Production of lentiviral particle

HEK293T cells were transfected with envelope plasmid (pMD2.G), packaging plasmid (psPAX2) and transfer plasmid (pSMPP-mCherry-hTRIM21 containing mCherry-hTRIM21) in the ratio pMD2.G:psPAX2: transfer plasmid of 1:1.84:2.1. After 16 h of transfection, cells were checked for syncytia formation and the transfection medium was

replaced with standard complete DMEM medium. After 48–72 h of transfection, the lentiviral particles were harvested by collecting the supernatant which was then centrifuged at 300$g$ for 5 min at 4°C. The lentiviral medium was sterile-filtered through 0.45 $\mu$m filters to remove dead cells before storage at −80°C. DNase digest was performed to remove any remaining plasmid DNA used for transfection by mixing 1 $\mu$g/ml DNase and 1 mM MgCl$_2$ gently to the lentiviral supernatant and incubating the mix at 37°C for 20 min.

## Transduction of cells using lentiviral vectors

A549 WT and A549 FLAG-FAT10 cells were transduced with the lentivirus particles carrying mCherry-hTRIM21 expression vector at a multiplicity of infection of 50 (MOI: 50) and selected using puromycin (10 $\mu$g/ml) from 48 h post-transduction. Single cell sorting of mCherry-TRIM21 expressing cells was performed using BD FACS AriaTM Ilu (BDBiosciences). Expression of mCherry-TRIM21 was confirmed by Western blot analysis and flow cytometry using BD FACS Lyric (BD Biosciences).

## Induction of endogenous FAT10

Endogenous expression of FAT10 in A549 cells was induced by co-stimulation with pro-inflammatory cytokines TNF (600 U/ml) and IFN$\gamma$ (300 U/ml) (both from Peprotech GmbH) for 24 h, as described earlier (Aichem et al, 2019a).

## Generation of CRISPR knockout cells

Generation of HEK293 UBA6 KO and USE1 KO has been described before (Aichem et al, 2018; 2019b). A549 FAT10 KO cells were generated by transfection of A549 cells with pCMV-Cas9-GFP containing FAT10-specific gRNA (Sigma-Aldrich). Single cells expressing high GFP were sorted using BD FACS AriaTM Ilu (BD Biosciences). A549 and A549 FLAG-FAT10 TRIM21 KO cells were generated by transfection of A549 and A549 FLAG-FAT10, respectively, with pSpCAS9 wt (BB)-2A-GFP targeting human TRIM21. Cells expressing GFP were single cell sorted using BD FACS AriaTM Ilu (BD Biosciences). The desired KO was confirmed using Western blot analysis.

## Recombinant proteins and in vitro FAT10ylation assay

Purification of recombinant FAT10 and FAT10-AV was described earlier (Aichem et al, 2019a). Recombinant FLAG-UBA6 was purchased from Enzo Life Sciences and human C-Myc-DDK-TRIM21 was purchased from Origene (#TP302088). To investigate FAT10 activation by FLAG-UBA6 and its conjugation with C-Myc-DDK-TRIM21, 0.54 $\mu$g FLAG-UBA6, 3 $\mu$g FAT10 or FAT10-AV, and 1 $\mu$g of C-Myc-DDK-TRIM21 were mixed in 20 $\mu$l 1x in vitro buffer (20 mM Tris–HCl, pH 7.6, 50 mM NaCl, 10 mM MgCl$_2$, 4 mM ATP, and 0.1 mM DTT, 1x protease inhibitor mix [Roche], with or without 4 mM ATP) and incubated for 60 min at 37°C. Reaction was stopped by adding 4x SDS sample buffer supplemented with 4% 2-ME and boiling.

## ELISA

A549 cells were infected with IAV strain A/Regensburg/D6/09 (H1N1pdm09, RB1) at an MOI of 1 in a six well dish. After 24 h, supernatants and cell pellets were collected for IFN$\beta$ ELISA and Western blot analysis, respectively. A classical sandwich-ELISA for IFN$\beta$ was performed according to the manufacturer's protocol (DY814-05; R&D System) with the supernatants of the infected cells. Uninfected cells were used as negative control. The absorbance was measured using a TECAN Infinite M200 pro plate reader.

## Real-time PCR

mRNA was purified using the RNeasy Plus Mini Kit (QIAGEN) according to the manufacturer's instructions. cDNA synthesis from 1 $\mu$g of total mRNA was performed using the Reverse Transcription System (Biozym). Real-time PCR was performed using the "Blue S'Green qPCR Kit" (Biozym). TOptical Gradient 96 Real-Time PCR Thermocycler and the qPCRsoft V3.1 software (both from Analytik) was used for analysis.

The primers used to measure *fat10* expression were: forward 5′-GGGATTGACAAGGAAACCACTA-3′; reverse 5′-TTCACAACCTGCTTCTTAGGG-3′. Expression of *fat10* was normalized to *rpl13a* which served as a house keeping gene. The primers for mouse *rpl13a* were forward 5′-CTACAGAAACAAGTTGAAGTACCTG-3′; reverse 5′-ATGCCGTCAAA-CACCTTGAG-3′. The primers used to measure human *ifnβ* were forward 5′-CAGCAGTTCCAGAAGGAGGA -3′; reverse 5′-AGCCAG-GAGGTTCTCAACAA-3′. Human *rpl13a* was used as a house keeping gene to normalize the expression of *ifnβ*. Primers for human *rpl13a* were forward 5′- CTACAGAAACAAGTTGAAGTACCTG -3′; reverse 5′-ATGCCGTCAAACACCTTGAG -3′. Results are shown as $2^{-\Delta\Delta Ct}$ values.

## Confocal microscopy

A549 cells were seeded on sterile glass cover slips in 12-well plates to a confluency of 25% and cultured at 37°C and 5% CO$_2$. At the following day, cells were infected with IAV at an MOI of 1. After 1 h, medium was replaced with fresh DMEM medium, TNF/IFN$\gamma$ were added for endogenous FAT10 induction, as described above. After 24 h, cells were washed with PBS and fixed with 4% formaldehyde in PBS for 10 min. Cells were washed twice in PBS buffer (PBS, 3% BSA) and permeabilized for 5 min with 0.2% Triton X-100/PBS. After two washing steps, A549 cells stably expressing GFP and/or mCherry were stained with DAPI (ab228549, 1:1,000; Abcam) and directly mounted on glass slides using Fluoromount-G mounting media (0100-01; Southern biotech). To visualize endogenous FAT10, TRIM21, and M1 protein of IAV, cells were incubated with primary antibodies for 2 h, followed by three time washing steps with PBS and incubation with fluorophore-labelled secondary antibodies for 2 h. Cells were washed with PBS thrice and nuclear staining was performed using DAPI (ab228549, 1:1,000; Abcam). Images were acquired with a 25x LDLCI-Plan Apochromat oil objective using a Zeiss LSM880 confocal microscope. The following primary antibodies were used: mouse anti-FAT10 4F1 ([Aichem et al, 2010], 1:1,000; Enzo Life Science), rabbit anti-TRIM21 (ab91423, 1:2,000; Abcam), mouse anti-M1 (ab22395, 1:1,000; Abcam). Following secondary antibodies were

used for the immunofluorescence: goat anti-mouse Alexa Fluor 647 (A-11019, 1:1,000; Thermo Fisher Scientific) and goat anti-rabbit Alexa Fluor 488 (A-11070, 1:1,000; Thermo Fisher Scientific).

## Data Availability

This study includes no data deposited in external repositories. Original, uncropped, and unprocessed scans of all gels used in the figures are shown in the Source Data. Supplementary data for this article are available.

## Supplementary Information

## Acknowledgements

We gratefully acknowledge deceased Prof. Dr. Marcus Groettrup for scientific support and discussions and for acquiring funding resources and we thank PD Dr. Gunter Schmidtke for scientific discussions. This work was supported by the Swiss State Secretariat for Education, Research and Innovation, the Deutsche Forschungsgemeinschaft (DFG) Collaborative Research Center CRC969 (TP C01 and C09), and DFG GR 1517/25, the "Konstanzia Transition" and the "Ausschuss für Forschungsfragen" (AFF) at the University of Konstanz in Germany.

### Author Contributions

K Saxena: conceptualization, validation, investigation, methodology, and writing—original draft.
K Inholz: methodology.
M Basler: funding acquisition and methodology.
A Aichem: conceptualization, supervision, funding acquisition, methodology, project administration, and writing—review and editing.

### Conflict of Interest Statement

The authors declare that they have no conflict of interest.

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
