## [Reviewer comments · Life Science Alliance]

Life Science Alliance

FAT10 inhibits TRIM21 to downregulate antiviral type-I interferon secretion

Kritika Saxena, Katharina Inholz, Michael Basler, and Annette Aichem

DOI: <https://doi.org/10.26508/lsa.202402786>

Corresponding author(s): Annette Aichem, Biotechnology Institute Thurgau

Review Timeline:

Submission Date:	2024-04-22
Editorial Decision:	2024-05-24
Revision Received:	2024-06-19
Editorial Decision:	2024-06-25
Revision Received:	2024-06-27
Accepted:	2024-06-27

Transaction Report:

May 24, 2024

Re: Life Science Alliance manuscript #LSA-2024-02786

Dr. Annette Aichem
Biotechnology Institute Thurgau
Unterseestrasse 47
Kreuzlingen 8280
Switzerland

Dear Dr. Aichem,

Thank you for submitting your manuscript entitled "FAT10 inhibits TRIM21 to downregulate antiviral type-I interferon secretion" to Life Science Alliance. The manuscript was assessed by expert reviewers, whose comments are appended to this letter. We invite you to submit a revised manuscript addressing the Reviewer comments.

Thank you for this interesting contribution to Life Science Alliance. We are looking forward to receiving your revised manuscript.

Sincerely,

B. MANUSCRIPT ORGANIZATION AND FORMATTING:

Reviewer #1 (Comments to the Authors (Required)):

Summary

The authors showed that FAT10, a ubiquitin-like modifier, could covalently conjugate to TRIM21, an E3 ligase, and an IFN-I regulator and target TRIM21 for proteasome degradation. The C-terminal diglycine motif of FAT10 was found to be critical for interacting with the coiled-coil and PRYSPRY domains of TRIM21. Moreover, FAT10 was shown to reduce TRIM21 ubiquitination and suppress IFN- β production during IAV infection. FAT10 overexpression combined with TRIM21KO further accentuated the inhibitory effect on IFN- β production, suggesting that FAT10 has an additional role in the negative regulation of the innate immune response. Together, these results demonstrated that TRIM21 is a target for FAT10-mediated downregulation of antiviral type I IFN response.

Major points

The authors presented compelling evidence that TRIM21 is a novel FAT10 interacting protein, although the exact lysine residue for FAT10ylation remains to be determined. More importantly, they showed the significance of FAT10ylation of TRIM21, which led to the attenuation of TRIM21 ubiquitination, resulting in the degradation of the protein and decrease of IFN-I production upon IAV infection. Unfortunately, the phenotypes could not be observed in mice following LCMV infection.

Specific points

1. Line 71-75, the authors described the positive role of TRIM21 in type I IFN production. However, TRIM21 also has a negative role in regulating the innate immune response (for example, *Nat Immunol* 2013 14:172 and *J. Neuroinflammation* 2014 11:24). The authors need to include this information in the introduction.
2. Fig. 1E, the expression of the TRIM21 Δ coiled-coil mutant appeared to be lower than the rest of the constructs. Could it be the reason why the interactions with FAT10 were impaired? The authors may need to increase the expression levels or other ways to rule out this possibility.
3. Fig 2B, the effect of UBA6 rescue (left panel lane 5) for restoration of FAT10ylation on TRIM21 was less pronounced, most likely due to inefficient expression of this protein in UBA6KO cells (right panel lane 5). It is desirable to repeat the experiment with a better expression of UBA6.
4. Fig. 4A, the results showed that FAT10 conjugation to TRIM21 was independent of IAV infection, although IAV could increase FAT10ylation on TRIM21. Is FAT10ylation on TRIM21 constitutive, or is it dependent on viral infection-induced post-translational modification, such as the phosphorylation of its target protein? How to reconcile the results with the reduced TRIM21 in FAT10 expression only after IAV infection (Fig. 5A and 5C)? Moreover, the levels of TRIM21 seemed to increase after IAV infection, which is inconsistent with the results shown in Fig. 5A and 5C; why?
5. Fig. 5B and 5D, was the downregulation of IFN β by FAT10 at transcriptional or post-transcriptional levels? How about the RNA levels of IFN β ? How about the IAV titers in the infected cells? Were IAV titers increased in those cells with decreased IFN β production?
6. Fig. 6, The effects of FAT10ylation on TRIM21, including degradation and suppression of ubiquitination, should be illustrated in the schematic diagram.
7. Fig. S3A How about the IAV levels as indicated by M1 staining?

Reviewer #2 (Comments to the Authors (Required)):

Saxena et al address the role of FAT10 is modulating type I IFN production following viral infection. They find that FAT10 becomes covalently linked to TRIM21 which targets it for degradation by the proteasome, leading to a decrease in type I IFN production. This constitutes a novel mechanism through which FAT10 functions.

The experiments that are described were well chosen to address the proposed hypotheses. The data in the figures are clear and support their conclusions. The text is well written. This work would be a welcome addition to the literature.

The manuscript could be improved by considering the following minor suggestions:

1. Explain to the readers why there are two FLAG bands in Figs. 1A and B.
2. The authors explain that DDK and FLAG are the same entity. It would be helpful for the reader if only one of them were used to refer to that protein tag.
3. In the manuscript the authors refer to E1 and E2 enzymes without other explanation. It would be appropriate to give a short background explaining how the E3 ligase works in conjunction with the E1-ubiquitin activating enzyme and the E2 ubiquitinating conjugating enzyme.
4. The team used different cells to test this system. They should explain why the chosen cell lines are appropriate for these studies, and why different cells were used for different assays.
5. In line 233, the authors refer to a typical FAT10ylated protein smear. It would be helpful to explain what this smear represents and why it is typical.
6. In line 238, the authors refer to semi-endogenous cellular conditions. They should explain what semi-endogenous means, or choose a better term.
7. The reader may wonder whether the perceived reduction in TRIM21 ubiquitination in Fig. 4D just reflects lower levels of TRIM21. Some explanation may be helpful.

Point-by-point reply for manuscript #LSA-2024-02786

We are grateful to both reviewers for their constructive criticism. We believe that our manuscript has benefited greatly from this revision. In the point-by-point reply below, we have answered all questions, asked by our reviewers and hope that the new experiments and text changes will now satisfy their requests.

For an easier orientation, we have additionally uploaded a manuscript version in which all text changes are highlighted in yellow.

Reviewer #1 (Comments to the Authors):

Summary

The authors showed that FAT10, a ubiquitin-like modifier, could covalently conjugate to TRIM21, an E3 ligase, and an IFN-I regulator and target TRIM21 for proteasome degradation. The C-terminal diglycine motif of FAT10 was found to be critical for interacting with the coiled-coil and PRYSPRY domains of TRIM21. Moreover, FAT10 was shown to reduce TRIM21 ubiquitination and suppress IFN- β production during IAV infection. FAT10 overexpression combined with TRIM21KO further accentuated the inhibitory effect on IFN- β production, suggesting that FAT10 has an additional role in the negative regulation of the innate immune response. Together, these results demonstrated that TRIM21 is a target for FAT10-mediated downregulation of antiviral type I IFN response.

Major points

The authors presented compelling evidence that TRIM21 is a novel FAT10 interacting protein, although the exact lysine residue for FAT10ylation remains to be determined. More importantly, they showed the significance of FAT10ylation of TRIM21, which led to the attenuation of TRIM21 ubiquitination, resulting in the degradation of the protein and decrease of IFN-I production upon IAV infection. Unfortunately, the phenotypes could not be observed in mice following LCMV infection.

Specific points

1. Line 71-75, the authors described the positive role of TRIM21 in type I IFN production. However, TRIM21 also has a negative role in regulating the innate immune response (for example, Nat Immunol 2013 14:172 and J. Neuroinflammation 2014 11:24). The authors need to include this information in the introduction.

We completely agree with this comment. We have now included this information about TRIM21 in the introduction:

Lines 70ff: "TRIM21 (also known as Ro52) has been shown to play an important role in the innate immune response, particularly during viral infections (Oke & Wahren-Herlenius, 2012) and its expression is induced by type-I IFNs (Bottermann & James, 2018). However, the exact role of TRIM21 in the antiviral immune response is not yet completely understood since there are contradictory studies showing positive as well as negative effects of TRIM21. For example, in the absence of TRIM21, the innate immune response towards an RNA virus is severely disabled (Foss et al, 2019) and therefore TRIM21 has been classified as a positive regulator of type-I IFN secretion, stabilizing or activating several molecules of the type-I IFN cascade (Li et al, 2023). On the other hand, TRIM21 has been described to be a negative regulator of the innate immune response following infection of myeloid dendritic cells and monocytes with DNA viruses, and upon infection of a human microglial cell line (CHME3) with single-stranded RNA virus JEV (Japanese encephalitis virus) (Zhang et al, 2013; Manocha

et al, 2014), which might point to a cell- and/or virus type-specific function of TRIM21 in the antiviral immune response.”

2. Fig. 1E, the expression of the TRIM21 Δ coiled-coil mutant appeared to be lower than the rest of the constructs. Could it be the reason why the interactions with FAT10 were impaired? The authors may need to increase the expression levels or other ways to rule out this possibility.

We understand the concern of our reviewer and have now repeated this experiment to increase the expression of the TRIM21 Δ coiled-coil mutant. In new Figure 1E we show now equal amounts of all TRIM21 truncation variants in the load, as well as in the IP (IP: FLAG, IB: FLAG). Since there is still no interaction of this variant with HA-FAT10 detectable (IB: FLAG, IB: HA), our initial statement that this TRIM21 domain is involved in the FAT10 interaction holds true.

3. Fig 2B, the effect of UBA6 rescue (left panel lane 5) for restoration of FAT10ylation on TRIM21 was less pronounced, most likely due to inefficient expression of this protein in UBA6KO cells (right panel lane 5). It is desirable to repeat the experiment with a better expression of UBA6.

We agree with our reviewer that the signal of the TRIM21-FAT10 conjugate in the immunoprecipitation in lane 5 is not very strong upon reconstitution of UBA6-ko cells with a UBA6 expression plasmid. We have made the observation that overexpression of any protein in UBA6-ko cells is always lower as compared to wild type cells. Therefore, we have repeated this experiment and have used a higher volume of cell lysate for the samples with UBA6-ko cells to increase the signal for UBA6 (new Figure 2B). The signal of the TRIM21-FAT10 conjugate is now as strong as in HEK293 wild type cells, while still no signal is detected in UBA6 KO cells without UBA6 reconstitution (Fig. 2B, lanes 2, 3 and 5). We have added this information to the figure legend of Fig 2:

Line 803: “In case of UBA6 KO cells, a 4x times amount of the cell lysate was used for immunoprecipitation, since overexpression of proteins is always low in this cell line.”

4. Fig. 4A, the results showed that FAT10 conjugation to TRIM21 was independent of IAV infection, although IAV could increase FAT10ylation on TRIM21. Is FAT10ylation on TRIM21 constitutive, or is it dependent on viral infection-induced post-translational modification, such as the phosphorylation of its target protein? How to reconcile the results with the reduced TRIM21 in FAT10 expression only after IAV infection (Fig. 5A and 5C)? Moreover, the levels of TRIM21 seemed to increase after IAV infection, which is inconsistent with the results shown in Fig. 5A and 5C; why?

We thank our reviewer for this question. For reasons of convenience we have split our answers to this question:

“Fig. 4A, the results showed that FAT10 conjugation to TRIM21 was independent of IAV infection, although IAV could increase FAT10ylation on TRIM21.”

In Fig. 4A it looks like an increase in the amount of the TRIM21-FAT10 conjugate upon IAV infection, however, this is because IAV infection increases the expression of TRIM21. Moreover, in this sample also slightly more FLAG-FAT10 was present (load, IB: FLAG, lane 3), therefore we also see more of the conjugate. Hence, we strongly assume that the enhanced amount of the TRIM21-FAT10 conjugate upon IAV infection is due to the increased TRIM21 and FAT10 amount and not due to a direct IAV effect on conjugation.

“Is FAT10ylation on TRIM21 constitutive, or is it dependent on viral infection-induced post-translational modification, such as the phosphorylation of its target protein?”

We see FAT10ylation of TRIM21 as soon as FAT10 and/or TRIM21 are overexpressed, either in the presence or in the absence of IAV infection (as for example shown in Fig. 1 or 4A). We therefore suggest that FAT10ylation of TRIM21 must rather be constitutive and not dependent on IAV-induced post-translational modifications. We of course cannot exclude, that TRIM21 is phosphorylated or modified by certain post-translational mechanisms upon viral infection, however, such modifications seem not to play role for FAT10ylation of TRIM21.

“How to reconcile the results with the reduced TRIM21 in FAT10 expression only after IAV infection (Fig. 5A and 5C)? Moreover, the levels of TRIM21 seemed to increase after IAV infection, which is inconsistent with the results shown in Fig. 5A and 5C; why?”

In Fig 4A we investigated the interaction of FAT10 and TRIM21 in a Ni-IDA pull-down experiment. We observed that upon IAV infection more TRIM21 could be pulled down. However, TRIM21 was only analyzed in the presence of FAT10. IAV infected wild type cells without FAT10 expression were not required for this experiment. In contrast, in Fig. 5A and 5C, TRIM21 expression was analyzed in the presence or absence of FAT10 upon IAV infection. Compared to IAV infected cells not expressing FAT10 (wild type cells), cells expressing FAT10 and infected with IAV showed a reduced amount of TRIM21. Hence, Fig. 4A and Fig. 5A, C cannot directly be compared, since in Fig. 4A TRIM21 was only analyzed in the presence of FAT10.

5. Fig. 5B and 5D, was the downregulation of IFN β by FAT10 at transcriptional or post-transcriptional levels? How about the RNA levels of IFN β ? How about the IAV titers in the infected cells? Were IAV titers increased in those cells with decreased IFN β production?

We think that this is a very interesting comment. We have now added additional data to new supplementary Figure S2. As can be seen in new Fig S2C and S2G, IAV titers did not significantly change in the different cell lines, showing that the diminished amount of produced IFN β did not result from a milder infection of these cells. Moreover, we did not see an increase in IAV titers in cells secreting lower levels of IFN β . To investigate the effect of reduced IFN β on IAV titers cells should be cultured for a longer time period.

In addition, the amount of IFN β mRNA in the respective cells (shown in new Figure S2D and S2H) clearly correlated with the secreted IFN β protein levels shown in the ELISA data in Fig. 5B and D, suggesting a transcriptional regulation of FAT10-mediated downregulation of type-I interferon secretion. Moreover, we have now also included ELISA data showing that no IFN β protein was detectable in uninfected cells (new Fig S2B and S2F). We have added these new results to the main text:

Line 312: “While uninfected cells did not produce detectable amounts of IFN β (Fig S2B),...”

Lines 319ff: “As a confirmation of the obtained results, IAV titers were determined to ensure that all cell types were infected with the same virus load (Fig S2C). Real-time PCR further confirmed that the observed downregulation of secreted IFN β protein correlated with a reduced IFN β mRNA expression and thus was due to a reduced transcription and not due to a degradation of the IFN β protein (Fig. S2D).”

Line 338: “Uninfected cells did again not express detectable amounts of IFN β as measured by ELISA (Fig S2F).”

Lines 343ff: “Also under these experimental conditions, viral titers were equal in all infected cells (Fig S2G) and the IFN β mRNA levels (Fig S2H) correlated with the secreted IFN β protein amounts shown in Fig 5D, confirming that the observed down-regulation of IFN β was mediated on the transcriptional level.”

6. Fig. 6, The effects of FAT10ylation on TRIM21, including degradation and suppression of ubiquitination, should be illustrated in the schematic diagram.

We have now changed the illustration shown in Fig 6 according to the reviewer's suggestions.

7. Fig. S3A How about the IAV levels as indicated by M1 staining?

In the original figure, we did not include the M1 staining for technical reasons. Both, the antibody directed against FAT10 as well as the antibody directed against M1, have been generated in mice. Therefore, we could not use them for a co-staining. However, the M1 staining was always performed in parallel with the same batch of cells, infected and induced at the same day and same time point. We have now included these data in new Fig. S3B, showing the corresponding M1 staining in the same batch of A549 cells as shown in Fig. S3A.

Reviewer #2 (Comments to the Authors (Required)):

Saxena et al address the role of FAT10 is modulating type I IFN production following viral infection. They find that FAT10 becomes covalently linked to TRIM21 which targets it for degradation by the proteasome, leading to a decrease in type I IFN production. This constitutes a novel mechanism through which FAT10 functions.

The experiments that are described were well chosen to address the proposed hypotheses. The data in the figures are clear and support their conclusions. The text is well written. This work would be a welcome addition to the literature.

The manuscript could be improved by considering the following minor suggestions:

1. Explain to the readers why there are two FLAG bands in Figs. 1A and B.

We thank our reviewer for this comment. Indeed we see such a double band of TRIM21 only under overexpressing conditions when expressing the myc-DDK-tagged variant of TRIM21, or in our *in vitro* experiments when using purified myc-DDK-TRIM21 from HEK293 cells (the purified myc-DDK-TRIM21 protein was purchased from Origene). Since we see this double band only in case of the usage of the myc-DDK tag but never when detecting endogenous TRIM21, we think that we can exclude, that this is a post-translational modification of TRIM21 itself, however, we cannot exclude such a modification of the myc-DDK tag. Alternatively, the second band might also be a specific degradation product, which cannot be FAT10ylated anymore. We mention this now in the main text:

Lines 162ff: "Of note, myc-DDK-TRIM21 always appeared as a double band as compared to endogenous TRIM21 as shown in experiments in Fig 4 and 5. We therefore suggest that the second band might represent rather a post-translational modification of the myc-DDK-tag than of TRIM21 itself. Alternatively, it might also represent a specific degradation product of myc-DDK-TRIM21, which cannot be FAT10ylated anymore."

2. The authors explain that DDK and FLAG are the same entity. It would be helpful for the reader if only one of them were used to refer to that protein tag.

We understand that the usage of both names might be confusing. However, since the plasmid purchased from the company Origene is originally named pCMV6-myc-DDK-TRIM21, and since the

company SIGMA uses the name “FLAG” as a brand name for the DDK-tag, we decided not to change the labeling. Although we had this information already included in the main text (Line 153f, “the FLAG tag is also referred to as DDK tag”), we have now additionally added a short explanation to all relevant Figure legends. For Fig. 1 we already had the following information included in the text “HEK293T cells were transiently transfected with expression constructs for HA-FAT10 and Myc-DDK-TRIM21. After 24 hours, cells were harvested and lysed. Cleared lysate was subjected to immunoprecipitation (IP) using FLAG M2 affinity gel, which specifically recognizes the DDK (FLAG) tag.” In the Figure legends for Fig. 2 and Fig. 3 we have now included this information, as well. In our opinion these additional explanations will be helpful to avoid any confusion.

3. In the manuscript the authors refer to E1 and E2 enzymes without other explanation. It would be appropriate to give a short background explaining how the E3 ligase works in conjunction with the E1-ubiquitin activating enzyme and the E2 ubiquitinating conjugating enzyme.

We agree and have added the following section to the introduction:

Lines 93ff: “Briefly, during the process of ubiquitination, ubiquitin binds to the adenylation domain of one of the two known E1 activating enzymes, UBE1 and UBA6 (Ciechanover et al, 1981; Jin et al, 2007; Pelzer et al, 2007), where it becomes adenylated at its C-terminal glycine residue. The activated modifier is then transferred onto the active site-cysteine of the same E1 enzyme to form a thioester bond. In the next step, it is transferred to the active site cysteine of a cognate E2 conjugating enzyme by a transthioesterification reaction, likewise forming a thioester bond. Finally, different classes of ubiquitin ligases (E3s) catalyze the isopeptide linkage of ubiquitin to the ϵ -amino-group of an internal lysine residue of a substrate protein (Kerscher et al, 2006; Finley, 2009).”

4. The team used different cells to test this system. They should explain why the chosen cell lines are appropriate for these studies, and why different cells were used for different assays.

We explain now that we have used HEK293 cells for all experiments where we overexpressed myc-DDK-TRIM21 and tagged FAT10, and the lung epithelial cell line A549 for all experiments where we applied IAV infection:

Line 151f: “HEK293T cells were used for all overexpressing experiments, since these cells are easy to transfect and because in these cells, FAT10 expression does not induce apoptosis, as for example shown in case of HeLa cells (Raasi et al, 2001).”

Line 237f: A549 cells were used instead of HEK293T cells, since these cells are robustly infected with IAV and are capable to produce and to secrete IFN β , thus, mimicking *in vivo* conditions.”

5. In line 233, the authors refer to a typical FAT10ylated protein smear. It would be helpful to explain what this smear represents and why it is typical.

We agree with this comment. The term “smear” is often used to describe the bulk of ubiquitin or ubiquitin-like modifier conjugates. Since these are many different proteins of different sizes, they appear as a smear instead of distinct bands. We have changed the sentence as follows:

Lines 266f: “A typical smear of FAT10ylated proteins was observed in A549 FLAG-FAT10 pull-down samples when immunoblotted with an antibody reactive to FLAG (Fig 4A, Ni-IDA-PD, IB: FLAG, lanes 2 and 3), validating an efficient pull-down of FLAG-FAT10 using Ni-IDA agarose beads and representing the bulk of all FAT10ylated proteins having different molecular weights.”

6. In line 238, the authors refer to semi-endogenous cellular conditions. They should explain what semi-endogenous means, or choose a better term.

Semi-endogenous means that one of the two proteins is expressed endogenously, while the second protein is overexpressed. We have explained the term now as follows:

Lines 269f: “Taken together, we could show that TRIM21 is modified with FAT10 under semi-endogenous cellular conditions, meaning by investigating the modification of endogenous TRIM21 with overexpressed FAT10, using Ni-IDA pull-down of 6xHIS-3xFLAG-FAT10 in A549 cells, bypassing the need for the antibody-mediated immunoprecipitation.”

7. The reader may wonder whether the perceived reduction in TRIM21 ubiquitination in Fig. 4D just reflects lower levels of TRIM21. Some explanation may be helpful.

We agree that we see a slightly higher amount of TRIM21 in the IP: TRIM21, IB: TRIM21, when comparing lane 2 with lanes 1, 3, and 4. However, the difference in the amount of ubiquitinated TRIM21 in lane 4 as compared to lane 2 is much stronger than the slight decrease in immunoprecipitated TRIM21 in lane 4 as compared to lane 2 (IP: TRIM21, IB: TRIM21). This is also visible when the ubiquitination of TRIM21 in lane 1 and 3 is compared with that in lane 4, while in these three lanes, equal amounts of TRIM21 were immunoprecipitated. To clarify this we have added the following sentence:

Lines 293ff: “Of note, equal amounts of immunoprecipitated TRIM21 were observed in lanes 1, 3 and 4 and only slightly, IAV-mediated increased TRIM21 levels were observed in lane 2. Nevertheless, we saw a considerable reduction in ubiquitination of TRIM21 in cells overexpressing FLAG-FAT10 and infected with IAV (IP: TRIM21, IB: Ub, lane 4 versus lanes 1-3).”

June 25, 2024

RE: Life Science Alliance Manuscript #LSA-2024-02786R

Dr. Annette Aichem
Biotechnology Institute Thurgau
Unterseestrasse 47
Kreuzlingen 8280
Switzerland

Dear Dr. Aichem,

Thank you for submitting your revised manuscript entitled "FAT10 inhibits TRIM21 to downregulate antiviral type-I interferon secretion". We would be happy to publish your paper in Life Science Alliance pending final revisions necessary to meet our formatting guidelines.

- please be sure that the authorship listing and order is correct
- please move your main and supplementary figure legends to the main manuscript text after the references section
- we encourage you to revise the figure legend for Figure S5 such that the figure panels are introduced in alphabetical order
- please add a callout for Figure S5F to your main manuscript text
- Figure 6 could be uploaded as a Graphical Abstract rather than as a figure, but this is up to you

FIGURE CHECKS:

- it is hard to read the scale bars in Figure S3

A. FINAL FILES:

B. MANUSCRIPT ORGANIZATION AND FORMATTING:

Thank you for your attention to these final processing requirements. Please revise and format the manuscript and upload materials within 5 days.

Sincerely,

June 27, 2024

RE: Life Science Alliance Manuscript #LSA-2024-02786RR

Dr. Annette Aichem
Biotechnology Institute Thurgau
Unterseestrasse 47
Kreuzlingen 8280
Switzerland

Dear Dr. Aichem,

Thank you for submitting your Research Article entitled "FAT10 inhibits TRIM21 to downregulate antiviral type-I interferon secretion". It is a pleasure to let you know that your manuscript is now accepted for publication in Life Science Alliance. Congratulations on this interesting work.

DISTRIBUTION OF MATERIALS:

Again, congratulations on a very nice paper. I hope you found the review process to be constructive and are pleased with how the manuscript was handled editorially. We look forward to future exciting submissions from your lab.

Sincerely,
